**Seasonal and Interannual Variability of Melt-Season Albedo at Haig Glacier, Canadian Rocky Mountains**

Shawn J. Marshall[1,2] and Kristina Miller[1]
shawn.marshall@ucalgary.ca

[1] Department of Geography, University of Calgary, Calgary, Alberta, T2N 1N4 Canada
[2] Environment and Climate Change Canada, Gatineau, Quebec, Canada

**Abstract**

In situ observations of summer (June through August, or JJA) albedo are presented for the period 2002-2017 from Haig Glacier in the Canadian Rocky Mountains. The observations provide insight into the seasonal evolution and interannual variability of snow and ice albedo, including the effects of summer snowfall, the decay of snow albedo through the melt season, and the potential short-term impacts of regional wildfire activity on glacier albedo reductions. Mean JJA albedo ($\pm$ 1$\sigma$) recorded at an automatic weather station in the upper ablation zone of the glacier was $\alpha_s = 0.55 \pm 0.07$ over this period, with no evidence of long-term trends in surface albedo. Each summer the surface conditions at the weather station undergo a transition from a dry, reflective spring snowpack ($\alpha_s$ ~0.8), to a wet, homogeneous mid-summer snowpack ($\alpha_s$ ~0.5), to exposed, impurity-rich glacier ice, with a measured albedo of $0.21 \pm 0.06$ over the study period. The ice albedo drops to ~0.12 during years of intense regional wildfire activity such as 2003 and 2017, but it recovers from this in subsequent years. This seasonal albedo decline is well-simulated through a parameterization of snow-albedo decay based on cumulative positive degree days, but the parameterization does not capture the impact of summer snowfall events, which cause transient increases in albedo and significantly reduce glacier melt. We introduce this effect through a stochastic parameterization of summer precipitation events within a surface energy balance model. The amount of precipitation and the date of snowfall are randomly selected for each model realization, based on a pre-defined number of summer snow events. This stochastic parameterization provides an improved representation of the mean summer albedo and mass balance at Haig Glacier. We also suggest modifications to conventional degree-day melt factors to better capture the effects of seasonal albedo evolution in temperature index or positive-degree day melt models on mountain glaciers. Climate, hydrology, or glacier mass balance models that use these methods typically use a binary rather than continuum approach to prescribing melt factors, with one melt factor for snow and one for ice. As alternatives, monthly melt factors effectively capture the seasonal albedo evolution or melt factors can be estimated as a function of the albedo, where this data is available.

**1. Introduction and Aims**

Melting of snow and ice is driven by the net radiative, turbulent, and conductive energy fluxes at the surface. Observations indicate the primary role of net radiation in driving snow and ice melt on mid-latitude glaciers (Greuell and Smeets 2001; Klok and Oerlemans 2002; Hock 2005; Marshall 2014). At Haig Glacier in the Canadian Rocky Mountains, net radiation provided ~80% of the net energy that was available for melt in the summer months (June through August, or JJA) from 2002-2012 (Marshall, 2014). Net radiation is dominated by net shortwave radiation in the summer melt season. For instance, in the Haig Glacier study noted above (Marshall, 2014), the linear correlation coefficient between daily mean values of net energy, $Q_N$, and net shortwave radiation is 0.84, compared with $-0.20$ for the net longwave radiation. The correlation coefficient between daily values of $Q_N$ and albedo is $-0.81$ in this dataset ($N = 1012$). Variations in surface albedo therefore exert a strong control on the surface energy balance and available melt energy (e.g., Reijmer et al., 1999; Ming et al., 2015; Ebrahimi and Marshall, 2016).

Mountain glaciers experience strong seasonal albedo variations, from ~0.9 for fresh, dry snow (i.e., the spring snowpack, at the start of the melt season), to ~0.5 for aged, wet snow or firn in mid-summer, to as low as 0.1 for impurity-rich glacier ice that is exposed after the seasonal snow has melted (Cuffey and Paterson, 2010). Albedo reductions through the melt season are due to recrystallization to larger, rounded grains, liquid water content in the snow, and increasing concentrations of impurities (Warren and Wiscombe 1980; Wiscombe and Warren, 1980; Marshall and Oglesby 1992; Conway et al., 1996; Gardner and Sharp, 2010).

A representation of seasonal albedo evolution is therefore important to accurate modelling of glacier melt (e.g., Brock et al., 2000; Klok and Oerlemans 2004). Where direct measurements of albedo are not available, the decrease in supraglacial snow albedo through the melt season is commonly parameterized as a function of snow depth and age (Wigmosta et al., 1994; Oerlemans and Knap, 1998; Klok and Oerlemans 2004). Alternatively, seasonal albedo decline can be based on a proxy for cumulative melting, such as cumulative positive degree days (*PDD*) or temperatures above 0°C (e.g., Brock et al., 2000; Bougamont et al., 2005; Hirose and Marshall, 2013). Seasonal snow typically melts away by mid- to late-summer on mid-latitude glaciers, exposing low-albedo ice or firn. The general trend of declining albedo through the summer melt season can be interrupted by snowfall events that temporarily increase surface albedo to fresh-snow values of ~0.9 (Oerlemans and Klok, 2003). High albedo values typically persist for a few hours to a few days, before the fresh snow melts away and the darker underlying-surface is re-exposed (Marshall, 2014).

A wide range of values for glacier ice albedo is reported in the literature, from ~0.1 to 0.6 (Bøggild et al., 2010; Cuffey and Paterson, 2010). Lower albedo values are mainly associated with high concentrations of particulate matter on the ice, which can accumulate over many melt seasons. Impurities on mountain glaciers are generally dominated by mineral dust (e.g., Oerlemans et al., 2009; Bühlmann, 2011; Nagorski et al., 2019), but include algae and cyanobacteria (e.g., Takeuchi et al., 2001, 2006; di Mauro et al., 2020; Williamson et al., 2019, 2020a), black carbon (soot) and other aerosols from incomplete combustion of fossil fuels, biomass burning, and forest fires (Ming et al., 2009; Keegan et al., 2014; de Magalhães Neto et

al., 2019; Nagorski et al., 2019), and other long-range contaminants, such as volcanic dust and heavy metals (e.g., Zdanowicz et al., 2014).

Elevated concentrations of black carbon on glacier surfaces due to increased industrial and wildfire activity have been raised as a concern for glacier mass balance, due to their direct impact on albedo and through melt-albedo feedbacks (Ming et al., 2009; Dumont et al., 2014; Keegan et al., 2014; Mernild et al., 2015; Tedesco et al., 2016; de Magalhães Neto et al., 2019). This is a positive feedback because melting concentrates impurities, further lowering the albedo and increasing melt rates. Similar concerns have been raised about melt-albedo feedbacks associated with algal activity on glaciers (Wientjes and Oerlemans, 2010; Stibal et al., 2017; Williamson et al., 2019; di Mauro et al., 2020). These processes are coupled, as microbial and algal activity require nutrients and meltwater, which increase in association with greater deposition and concentration of impurities, lower albedo, and longer melt seasons.

In addition, mineral dust deposition on glaciers can increase in association with glacier retreat (Oerlemans et al., 2009), due to exposure of fresh sources of material on the glacier margin as well as melt-concentration effects. Impurities on glacier surfaces are also transported and removed by rainfall and meltwater runoff, as both dissolved and suspended sediment. It is important to understand and separate these influences on glacier albedo, to document whether albedo is changing in recent decades, and to quantify the potential impact on glacier mass loss (Oerlemans et al., 2009; Dumont et al., 2014; Mernild et al., 2015; Williamson et al., 2020b).

This study examines the seasonal variability and multi-year trends in mean melt-season and glacier ice albedo from 14 years of surface albedo observations at Haig Glacier in the Canadian Rocky Mountains. We discuss the processes governing albedo fluctuations, including the potential impact of regional wildfire on surface darkening at our site and the influence of summer snowfall events, which introduce abrupt, transient increases in albedo. We quantify the impact of summer snowfall events on albedo and mass balance at Haig Glacier and introduce a simple stochastic parameterization of summer snowfalls to effectively capture their influence in models of glacier energy and mass balance.

A final aim of our study is to examine ways in which the seasonal albedo evolution on mountain glaciers can be implicitly included in temperature-index melt models, which remain widely used in glaciology (e.g., Marzeion et al., 2014; Clarke et al., 2015; Maussion et al., 2019; Jury et al., 2020). Temperature-index models of snow and ice melt can be necessary in mountain and polar environments where essential meteorological data are neither readily available nor easily modelled (Hock 2005; Fausto et al., 2009). Snow and ice melt are calculated using a melt factor which linearly relates the amount of melt to cumulative positive degree days (Braithwaite 1984) and potentially other influences, such as incoming shortwave radiation (Cazorzi and Dalla Fontana 1996; Hock, 1999). Melt factors are generally taken as constants for snow and for ice, with a higher value for ice due to its lower albedo. This binary treatment of the melt factor does not realistically represent the continuous nature of surface albedo values or the systematic evolution of albedo through the summer melt season. We explore parameterizations of the melt factor that better capture the effects of the seasonal albedo evolution on surface melt.

## 2 Study Site and Methods

### 2.1 Haig Glacier study site

Glaciological and meteorological measurements at Haig Glacier in the Canadian Rocky Mountains were initiated in August 2000 and are ongoing. Surface energy and mass balance characteristics of this site are summarized in Marshall (2014). Haig Glacier is the main outlet of a small icefield that straddles the North American continental divide between the provinces of British Columbia and Alberta (Figure 1). It flows southeastwards into Alberta, covering $2.62$ km$^2$ and spanning an elevation range from about 2520 m at the terminus to 2950 m in the upper accumulation area. Local geology is dominated by steeply dipping beds of limestone ($CaCO_3$) and dolostone ($MgCO_3$).

The region has mixed continental and maritime influences, being fed by Pacific air masses that transport moisture to the Canadian Rockies (Sinclair and Marshall, 2009). Snow surveys conducted on the glacier each May indicate a mean winter snowpack ($\pm$ 1$\sigma$) of $1.35 \pm 0.24$ m water equivalent (w.e.) on the glacier from 2002-2017 (Table 1). The standard deviation provides a measure of the interannual variability. Summer (June through August (JJA)) temperature on the glacier over the period of study averaged $5.3 \pm 0.8$°C from 2002-2017. The site has warm, sunny conditions in the summer months, driving average summer melt totals of $2.60 \pm 0.62$ m w.e over the study period. Annual mass balance at the site has been negative in every year of the study (Marshall, 2014; Pelto et al., 2019), with the glacier losing all of its winter snow in 9 of 16 years. Mean annual mass balance over the glacier from 2001-2017 was $-1.35 \pm 0.24$ m w.e., giving a cumulative mean thinning of about 24 m of ice over this period.

### 2.2 Field Measurements

Glacier albedo data and meteorological conditions are available from a Campbell Scientific automatic weather station (AWS) installed in the upper ablation zone of the glacier from the period 2001 to 2015. A second AWS was installed in the glacier forefield in 2001, and remains operational (Figure 1). Each AWS was equipped with sensors to measure temperature, humidity, wind speed and direction, incoming and outgoing longwave and shortwave radiation, rainfall, barometric pressure, and snow/ice surface height. AWS data were stored at 30-minute intervals, calculated from the average of 10-second measurements. The instrumentation is described in more detail in Marshall (2014). This study focuses on the albedo data, which were measured with two different radiation sensors over the study period. From 2001 to 2003, each AWS was deployed with upward- and downward-facing Kipp and Zonen CM6B pyranometers, integrating over the spectral range 0.31 to 2.80 μm, with a manufacturer-reported accuracy of within 5% for mean daily measurements (first class rating from the World Meteorological Organization). From July, 2003 to present we shifted to Kipp and Zonen CNR1 four-component radiometers, with a spectral range of 0.35-2.50 μm for the shortwave radiation. The manufacturer-reported accuracy of the CNR1 is 10% for mean daily net radiation.

The AWSs were maintained through an average of six visits per year, with sensors and dataloggers swapped out four times over the period 2001-2017, to be returned to the University of Calgary weather research station for calibration. The AWS observations are subject to manual quality control and any physically implausible data were removed from the analysis (i.e., values outside the normal range of conditions or a lack of variability, which occurs when a sensor is covered by snow). The AWS in the glacier forefield has been in place continuously but sensors can fail, the station was blown down once, and on two other occasions the station was buried by snow from the late spring through early summer. Hence there are occasional data gaps, but there is 92% data coverage for the summer period (JJA) from 2002-2017. The glacier AWS is more intermittent, due to the more difficult environment. It was maintained year-round from 2001 to 2008, but from 2009 to 2015 the station was set up only in the summer months. It was established at the same site each year. Quality-controlled data represent 79% of JJA days from 2002-2015 ($N_{JJA}$ = 1018).

Glacier albedo values presented in this study are restricted to the days with direct *in situ* values measured at the glacier AWS. For energy balance and melt modelling, missing meteorological data on the glacier are estimated from the off-glacier AWS, adjusted using monthly offsets calculated from the set of observations that are available from both sites, as explained in detail in Marshall (2014). As an example, for mean monthly temperatures $T_G(m)$ and $T_{FF}(m)$ on the glacier and in the glacier forefield, the mean monthly temperature offset is $\Delta T(m) = T_G(m) - T_{FF}(m)$. Where there is missing temperature data at the glacier AWS at time *t* during month *m*, it is gap-filled following $T_G(t) = T_{FF}(t) + \Delta T(m)$. For wind speed, incoming solar radiation, and specific humidity, the offset is applied through a ratio rather than a difference. For instance, missing wind-speed data are gap-filled following $v_G(t) = v_{FF}(t) \cdot v_G(m)/v_{FF}(m)$. If data are missing from both weather stations, gap-filling is based on the mean multi-year value recorded at the glacier AWS for a given day. This provides a complete meteorological forcing dataset for estimation of the summer energy and mass balance.

This manuscript focuses on the long-term albedo record at the glacier AWS site. We define the incoming and reflected shortwave radiation to be $Q_S^{\downarrow}$ and $Q_S^{\uparrow}$, respectively, with albedo $\alpha_s = Q_S^{\uparrow}/Q_S^{\downarrow}$. We are interested in the seasonal evolution and interannual variation of surface albedo and its influence on glacier mass balance, so we restrict our analysis to daily and longer timescales. Mean daily albedo is calculated from the integrated daily sum of incoming and outgoing shortwave radiation,

$$\alpha_{sd} = \frac{\int Q_S^{\uparrow} dt}{\int Q_S^{\downarrow} dt}. \tag{1}$$

Note that this gives different values from the average of instantaneous albedo measurements, as it weighs the albedo calculation to the middle of the day, when insolation is highest. This accurately reflects the amount of shortwave energy that is available for glacier melt. It also means that we do not consider zenith angle effects on surface albedo in this study.

Measurement uncertainty is reduced through daily averaging. The CM6B and CNR1 radiation sensors deployed at the glacier AWS have similar spectral ranges and manufacturer-reported accuracies of 5 to 10% for mean daily measurements. These values are conservative based on calibration studies indicating mean biases of 1 to 2% for total daily radiation measurements with these two sensors (Myers and Wilcox, 2009; Blonquist et al., 2009). Our sensors are not maintained on a daily basis, however, and are not respirated or heated, so we adopt an uncertainty of 5% for the mean daily incoming and outgoing radiation. Propagation of errors for division gives an uncertainty of 7% for the mean daily albedo (e.g., $0.60 \pm 0.04$ or $0.20 \pm 0.01$).

Sources of uncertainty in the albedo measurements include deviations from horizontality for measurements of incoming shortwave radiation, multiple reflections from the undulating glacier surface, reflected radiation from valley walls, and potential covering of upward-looking sensors during snow events, among other effects. The quality control measures identify obvious environmental corruption such as times when fresh snow is covering the sensors or excessive station leaning, and data from these days are omitted from the analysis. The glacier AWS is located near the glacier centreline, more than 400 m from the valley walls, with minimal impact from reflected radiation. The station experiences topographic shading within one hour of local sunrise and sunset during the summer melt season, but none through the day.

Additional data are included in this study from summer 2017, based on centreline surveys of albedo and chemical analyses of supraglacial snow and ice. These data, described in detail in Miller (2018), provide an additional spatial perspective on albedo variation, as well as insights about the provenance and concentration of impurities on the surface of Haig Glacier. Four surveys were conducted in July and August, 2017, at 33 sites on an altitude transect that approximates the glacier centreline (cf. Section 3 and Figure 7a). These measurements provide an indication of the variation in albedo with elevation on the glacier, its evolution over four different times through summer 2017, and the relation to supraglacial impurities.

For the spatial albedo surveys, measurements were only taken under clear-sky conditions and within three hours of local solar noon, to minimize the effects of diffuse radiation and high zenith angles. We used a Jaycar QM1582 portable pyranometer for these measurements, taking the average of three upward and three downward shortwave radiation measurements at each point. The sensor was held to the south at arms-length, at a height of ~1.1 m above the glacier surface for all measurements. The manufacturer-reported accuracy is 5%, but when measuring incoming shortwave radiation we observed considerable fluctuation in the reading, of order 10s of W m$^{-2}$. Fluctuations were within ±5% of the reading, but we take this as additive to the 5% error associated with instrumental accuracy, giving a total uncertainty of 10% for individual radiation measurements. With three measurements at each site, error propagation gives an uncertainty of 8% for the point albedo measurements. The spectral range of the handheld pyranometer is 0.3 to 4.0 μm. This extends further into the near infrared than the Kipp and Zonen instruments, and could bias the albedo to lower values. Caution is therefore needed in comparing values from this sensor with the AWS albedo data, although values are consistent at

the AWS site. We only intercompare data from this specific sensor, and not against the AWS-measured albedo values, to avoid the problem of differing spectral windows.

Surface snow and ice samples were collected at every third point and were melted, bottled, and analyzed for major ion and carbon concentrations. Samples were collected and melted in freezer bags and then transferred to 20-mL vials for transport to the University of Calgary, where they were analyzed in the Environmental Sciences program laboratory. For the ion analyses, 5 mL subsamples were run on a Metrohn 930 Compact Iron Chromatography (IC) Flex system, connected to Metrohm 858 autosampler. Concentrations are reported in mg $L^{-1}$, with precision and accuracy of 5% and a detection limit of 0.10 mg $L^{-1}$. Carbon concentrations of unfiltered glacier surface samples were analyzed on a Shimadzu TOC-V total organic carbon analyzer. For each sample, the machine measures the concentrations of total carbon and inorganic carbon (mg $L^{-1}$). The difference in these two values gives the concentration of organic carbon in the sample. Miller (2018) provides a complete summary of the chemical analyses.

**2.3 Energy Balance Model**

We use a distributed surface energy balance model to examine the influence of seasonal and interannual albedo variations on glacier mass balance and summer runoff for the period 2002-2017 (Ebrahimi and Marshall, 2016). The summer melt model is driven by 30-minute meteorological data from the glacier AWS. We carry out a survey of the winter snowpack each spring, typically in the second week of May, and use the winter mass balance data as an initial condition for the simulation of summer mass balance. Following Ebrahimi and Marshall (2016), the radiation data, parameterizations of the turbulent fluxes, and a subsurface model of snow/ice temperature and heat conduction provide the net surface energy flux, $Q_N$:

$$Q_N = Q_S^\downarrow(1 - \alpha_S) + Q_L^\downarrow - Q_L^\uparrow + Q_H + Q_E + Q_C, \tag{2}$$

where $Q_L^\downarrow, Q_L^\uparrow, Q_H, Q_E,$ and $Q_C$ represent incoming and outgoing longwave radiation, sensible and latent heat flux, and subsurface conductive energy flux, respectively. All energy fluxes have units of $W\,m^{-2}$ and are defined to be positive when they are sources of energy to the surface. Turbulent fluxes of sensible and latent energy are parameterized from a bulk aerodynamic method (Andreas, 2002) and surface temperature and conductive heat flux are modelled within a subsurface snow model that includes meltwater refreezing (Samimi and Marshall, 2017). When $Q_N$ is positive and $T_s = 0°C$, net energy is directed to melting, with melt rate

$$\dot{m} = \frac{Q_N}{\rho_w L_f}, \tag{3}$$

where $\rho_w = 1000$ kg $m^{-3}$ and $L_f = 3.35 \times 10^5$ J $kg^{-1}$ are the density and latent heat of fusion of water. Melt rates have units m w.e. $s^{-1}$. By integrating over the times when melt occurs (i.e., when $Q_N > 0$ and $T_s = 0°C$), one can calculate the total melt energy, $E_m$, over a period of time $\tau$, with units $J\,m^{-2}$. Melt over time $\tau$ is then calculated from

$$m(\tau) = \frac{E_m}{\rho_w L_f}. \tag{4}$$

This can be directly related to the classical positive degree day method (e.g., Braithwaite 1984), where snow or ice melt $m$ over a period of time $\tau$ is calculated from

$$m(\tau) = d_{s/i} \int_0^\tau \max(T, 0)\, dt, \tag{5}$$

where $d_{s/i}$ is the degree-day melt factor for snow or ice. This linearly relates the amount of melt to cumulative positive degree days (*PDD*) over time $\tau$. The integrand can also be modified to include other influences, such as the potential direct incoming shortwave radiation (Hock, 1999).

Eq. (5) is an empirical alternative to the physically-based approach in Eq. (4). It is useful because surface energy fluxes are uncertain in the absence of local AWS data, due to poorly-constrained meteorological input variables. Wind, humidity, cloud cover, and radiation fields are difficult to estimate in remote mountain terrain. Eq. (5) requires only temperature, which can be estimated via downscaling or interpolation of regional station data or climate model output. While appealing, it is recognized that this parameterization is over-simplified with respect to its transferability to other locations or times. For instance, there is no direct way to incorporate influences from meteorological variables other than temperature, and melt-albedo feedbacks are not physically represented where $d_s$ and $d_i$ are taken as constants.

Temperature-index models commonly approximate albedo effects by using different melt factors for ice and snow. Typical values are $d_i \sim$ 6-9 mm of water equivalent melt per degree day (mm w.e. $°C^{-1} d^{-1}$) and $d_s \sim$ 3-5 mm w.e. $°C^{-1} d^{-1}$ (Braithwaite, 1995; Jóhannesson, 1997; Hock, 2003; Casal et al., 2004; Shea et al., 2009). There is considerable local, regional, and temporal variability in the parameters chosen for different studies, with values sometimes twice as high as these, particularly for glacier ice (see Hock, 2003). Lefebre et al. (2002) also find a large spatial variation in melt factors, through modelling studies of melt patterns in Greenland. This variability is associated with differences in the energy balance and surface conditions that drive melt, much of which may be due to variations in surface albedo.

For regions where melting is the dominant process in glacier ablation (cf. Lett et al., 2019), one relatively simple way to improve on temperature-index models is to permit melt factors to vary in space and time, consistent with spatial and temporal variations in net energy and glacier albedo (Schreider et al., 1997; Arendt and Sharp, 1999). For melting over time $\tau$, one can combine Eqs. (4) and (5) to derive an expression for the melt factor at any location $(x, y)$:

$$d(x, y, \tau) = \frac{E_m(\tau)}{\rho_w L_f \int_0^\tau \max(T, 0)\, dt}. \tag{6}$$

Eq. (6) implicitly includes the seasonal evolution of surface albedo, as an important control on the melt energy, but numerous other meteorological influences are embedded in $E_m$, so there is not a direct relation between $d(t)$ and $\alpha_s(t)$. Because absorbed shortwave radiation is the dominant term driving ablation of mountain glaciers (Greuell and Smeets, 2001; Klok and

Oerlemans, 2002), including Haig Glacier (Marshall, 2014), one can expect that $E_m \propto 1/\alpha_S$. Moreover, melt energy is proportional to $PDD$, such that the numerator scales with the denominator in Eq. (6). Hence, it should be possible to develop a simple parameterization which includes the lead-order effects of surface albedo on the melt factor. We use Eq. (6) to calculate daily and monthly mean values of the melt factor, $d(t)$, at the Haig Glacier AWS site. A compilation of monthly mean values of $d$ and $\alpha_S$ informs a relation $d = f(\alpha_S)$ which represents the seasonal evolution of melt factors, if albedo is known or can be estimated (e.g., Williamson et al., 2020b). We consider different forms of this relation, but the simplest model is a linear parameterization $d(t) = a - b\alpha_S(t)$, for linear regression coefficients $a$ and $b$.

## 3. Results and Analysis

### 3.1 AWS Albedo Measurements, 2002-2015

Table 1 gives summary statistics for the observed Haig Glacier mass balance, net energy, and mean summer albedo at the glacier AWS site from 2002-2015. The winter mass balance is measured from snow surveys each May, and represents the snow accumulation from the end of the previous melt season (typically mid-September) through to May, averaged over the glacier surface. The summer mass balance is defined as the average glacier-wide mass change over the summer melt season, which typically runs from mid-May to mid-September. The specific dates vary from year to year and across the glacier. Positive degree day and melt totals are presented for the complete summer melt season, May through September, and temperatures, energy fluxes, and albedo values are given for the core summer months, June through August (JJA), when more than 80% of the melt occurs.

Table 2 reports the mean monthly values at the glacier AWS site for the 14-year record. The mean JJA surface albedo at this site is 0.55, with a marked decline through the summer months and a minimum of 0.38 in August (Figure 2a). Because the AWS is installed in the upper ablation zone of the glacier, seasonal snow gives way to bare glacier ice at some point in late summer. This is attended by a sharp drop in albedo, to the bare-ice value of $0.21 \pm 0.06$. Mean August albedo values represent an average of aged, wet snow and bare ice, with year-to-year variability associated with the timing of seasonal snow depletion. The average date of seasonal snow depletion at the AWS site is August 3, but it ranges from July 21 to August 20 over the study period. New snow accumulation at the AWS site ('winter snow') begins in September in most years, accounting for the albedo increase this month (Table 2). Persistent snow began to accumulate at the AWS site between August 30 and September 25 during our study period. On average, there are $25 \pm 10$ days with glacier ice exposed at the AWS site during the melt season. The site was selected because it is near the equilibrium mass balance point of the glacier: the elevation at which annual snow accumulation is equal to summer melt, for the glacier to be in balance. The observations of bare ice exposure are consistent with the persistent negative mass balance of the glacier over the period of observations.

Intermittent summer snow events also temporarily refresh the glacier surface (*e.g.*, Figure 2b), reducing the number of snow-free days on the glacier surface. The average melt-season albedo evolution at the site in Figure 2a averages out the impact of episodic summer snowfall events that refresh the snow or ice surface (*cf.* Figure 4d). As seen in Figure 2b, these cause an immediate increase in albedo to a fresh-snow value of ~0.9, followed by a decay back to the albedo of the underlying surface over the course of hours to a few days. Figure 3 provides a more detailed illustration of summer snowfall events over exposed glacier ice. This plot covers the period August 3-28, 2015, during which there were three distinct summer snow events, each of which increased the surface albedo for two to three days. The events can be predicted somewhat from the meteorological conditions, where temperature drops below 0°C and relative humidity reaches 100% (Figures 3a,b), but they are most clearly evident in the albedo record (Figure 3c). The accumulation of new snow is also apparent in the glacier surface height (SR50) data (Figure 3d), attended by an interruption in surface melting.

Even a modest amount of fresh snow has a strong albedo impact; there were roughly 4 cm of accumulation in the first two snow events and 8 cm for the third event in Figure 3. The latter event had a longer impact, roughly three days before the surface albedo returned to values typical of bare ice. Total surface ablation at the AWS site was 1.05 m over this 25-day period (Figure 3d), equivalent to about 0.95 m w.e. Based on the observed bare-ice vs. actual average albedo values over the 25-day period, 0.15 vs. 0.27, we calculate that the snow events reduced the average net energy by 24 W m$^{-2}$, equivalent to 0.16 m w.e. or a 17% reduction in melting over this period. Hence, the direct impact of summer snowfall on glacier mass balance is generally minor (estimated at ~0.03 m w.e. for the events in Figure 3), but the indirect impact through increased albedo and reduced melting is important.

Based on analysis of the SR50 and albedo data over the full study period, an average of $9.3 \pm 2.6$ ephemeral snowfall events per year occurred at the AWS site from May to September. This included $6.3 \pm 2.2$ summer (JJA) events. Our main criteria to identify summer snowfall events is a mean daily albedo increase of at least 0.15. This may not capture trace precipitation events that are too minor to be seen in either of the SR50 or albedo measurements (i.e., too ephemeral or not enough snow to mask the underlying surface), but these small events have limited impact on the mean summer albedo or mass balance.

**3.2 Relation Between Summer Albedo and Mass Balance**

The broader relations between glacier mass balance and albedo, summer snow events, and temperature at the Haig Glacier AWS site are summarized in Table 3. The bottom left portion of the table shows Pearson's linear correlation coefficients for monthly mean values of all variables ($N=70$), while the section above the diagonal shows correlation coefficients for the mean summer (JJA) values and the winter and annual mass balances ($N = 14$). Values that are not statistically significant at the 95% confidence level are shown in brackets. With the small sample size for the mean summer/mean annual variables, statistical significance requires $|r| > 0.53$. The greater number of months that are sampled permits a lower threshold for significance, $|r| > 0.23$.

Most variables in Table 3 are significantly correlated, with numerous interactions. The importance of albedo to melt and mass balance conditions is clear. Monthly mean albedo is highly correlated with monthly melt ($r = -0.88$) and net energy ($r = -0.84$), in addition to strong negative correlations with other melt indicators such as mean monthly temperature and *PDD* ($r = -0.73$). Monthly albedo values are also significantly correlated with the optimal monthly degree-day melt factor calculated from Eq. (6) ($r = -0.66$), which we discuss further in Section 4.2.

Correlation coefficients for the mean summer conditions are generally weaker. Mean melt-season albedo remains strongly correlated with summer and net mass balance, net energy, and temperature, but is only weakly associated with winter mass balance and total melt-season *PDD*. Winter mass balance is expected to impact the summer albedo, as deeper snowpacks take longer to ablate, delaying the transition to the low-albedo summer surface, but this is not as strong an influence as the summer melt conditions at Haig Glacier. In contrast, mean summer albedo is significantly correlated with the number of summer snow events ($r = 0.66$). Summer snow events have a significant overall influence on the summer and net mass balance ($r = -0.73$ and $r = 0.70$, respectively). Due to the compounding influences, mean summer albedo has a high association with net annual mass balance ($r = 0.86$). This is stronger than the correlation between mean summer temperature or *PDD* on net annual mass balance.

**3.3 Ice Albedo Values**

The progressive decline in glacier surface albedo through the melt season has been reported in many previous studies (e.g., Brock et al., 2000, Klok and Oerlemans, 2002, 2004). Within the seasonal snow, this can generally be related to cumulative melting, with its associated effects on snow grain size, liquid water content, and increasing concentration of impurities (Warren and Wiscombe, 1980). We discuss modelling of this seasonal evolution in Section 5. At Haig Glacier the wet, temperate July snowpack typically asymptotes to an albedo value of ~0.5, before surface albedo drops sharply to a value of ~0.2 once bare ice is exposed. Figure 4a visually captures this transition. For a composite of all bare-ice days in summer (JJA) in the 14-year record (i.e., days with no snow cover at the glacier AWS), the mean glacier albedo is $0.21 \pm 0.06$ ($N = 224$). Including the month of September, the number of bare-ice days increases to 272 and the mean ice albedo is $0.22 \pm 0.07$. Figure 5 plots the distribution of measured daily mean ice albedo values at the glacier AWS, ranging from 0.11 to 0.34.

Observed values for glacier ice albedo are in line with other mid-latitude glacier observations (e.g., Brock et al., 2000; Gerbaux et al., 2005; Naegeli et al., 2019) and the value of 0.2 recommended by Cuffey and Paterson (2010) for impurity-rich ice. Particulate concentrations are high in the old snow and glacier ice on Haig Glacier, and include a combination of mineral dust, black carbon, and organic material (see Section 4.4). Ice albedo values of 0.07 have been measured on the lower glacier in multiple years, in association with high impurity loads (Figure 4b,c). Indeed, during spatial albedo surveys, measured albedo is generally higher on the proglacial limestone than in the lower ablation zone (e.g., Figures 4b,c).

No part of Haig Glacier is considered to be debris-covered, where material covering the glacier is thick enough to insulate the ice surface from ablation; rather, supraglacial particulate matter takes the form of discrete particles or a thin (~1-mm) film, with considerable spatial heterogeneity and temporal variability in impurity concentrations. The heterogeneity may be associated with variable patterns of atmospheric deposition, flushing (cleansing of the glacier surface through rain events or meltwater runoff), and microbial/algal activity. Temporal changes in these processes may also explain some of the variability in ice albedo at the AWS site.

We sorted all bare-ice days into subsets of clear-sky and overcast conditions, based on incoming shortwave radiation measurements at the AWS (specifically, the ratio of the total daily and potential direct incoming solar radiation). The spectral reflectance of snow is dependent on the solar incidence angle (Hubley, 1955; Wiscombe and Warren, 1980), hence differs for direct vs. diffuse radiation. As a result, mean daily albedo values can be expected to be higher on cloudy days, when diffuse radiation is dominant (Cutler and Munro, 1996; Brock, 2004; Abermann et al., 2014). However, we found no difference between the mean ice albedo values for clear-sky and overcast days in our dataset (a mean value of 0.21 for each subset). Glacier ice albedo is less sensitive to the zenith angle than snow (Cutler and Munro, 1996), and this may be particularly true for impurity-rich glacier ice, due to isotropic absorption by impurities and liquid water on the glacier surface and in the intergranular interstices.

However, another phenomenon may contribute to some of the higher ice-albedo values recorded at our AWS site. Values above 0.3 are most common in September, after ephemeral snowfall events have melted away; these serve to increase albedo by 0.1 to 0.2 above the minimum seasonal values attained in August. We interpret this to be due to either residual, refrozen (i.e. superimposed) ice that is more reflective or because of meltwater flushing of some of the local impurities. The albedo record in Figure 3c, an example from summer 2015, illustrates this temporary increase in albedo in the days following fresh snowmelt, particularly for the third snow event on August 22. The storm snow was melted away by August 24 (Figure 3d), but the exposed ice albedo values remained above their early-August 'baseline' value of ~0.13 until August 28. Albedo values after this returned to the baseline, indicating that a potential crust of superimposed or flushed ice had been melted away. This pattern is typical of the albedo evolution following summer snow events.

Figure 6a plots mean daily ice albedo values through the summer melt season, based on the average of all available data from 2002 to 2015. There is no trend of ice albedo decrease through the summer melt season; hence, no evidence of increasing impurities that cause progressive darkening once the glacier surface ice is exposed. In contrast, ice albedo increases in late August and September, perhaps associated with the brightening influence of superimposed ice or meltwater flushing, as hypothesized above. Summer 2003 was an interesting exception, plotted in red in Figure 6a. Ice albedo declined through July and the first week of August in 2003, reaching a minimum of 0.11 (the lowest mean daily value on record at the Haig AWS) and remaining at ~0.13 until strong melting caused the AWS to lean beyond a condition that ensures

reliable data after August 22. Severe wildfire conditions in southwestern Canada that summer resulted in an evacuation order for the region in mid-August, so we were forced to leave the site and could not maintain the AWS. These same wildfire conditions may have resulted in deposition of soot and black carbon that produced the extremely low albedo values that summer.

Figure 6b plots the 14-year record of mean and minimum summer albedo at the site for the period 2002-2015. Mean melt-season values are shown for both May through September and JJA. There is interannual variability but not temporal trend to these or to the minimum values over the study period. Notably, the 2003 ice albedo minimum noted above (Figure 6a) was tied for the lowest in the AWS record, matched again in 2015. The ice at the AWS site had a moderately higher albedo in intervening years. The lack of a trend either during the melt season (Figure 6a) or over multiple years (Figure 6b) implies that impurities must be flushed at a rate similar to their concentration through melting, at least in the upper ablation zone.

**3.4 Albedo Transects and Snow/Ice Impurity Data**

To supplement the AWS albedo record from a single point on the glacier, we conducted spatial albedo surveys across the glacier during different seasons. In summer 2017 we completed four centrelines albedo surveys through July and August, in conjunction with collection of snow and ice samples to analyze the chemistry and concentration of particulate matter on the glacier surface. Figure 7 plots the location of the survey sites and the centreline albedo data from these four surveys, with summary data provided in Table 4.

The characteristic decrease in surface albedo on Haig Glacier over the summer melt season is evident in Figure 7b. For the initial survey on July 13, the glacier surface was still completely snow-covered, with a relatively uniform albedo typical of old, wet snow. The average albedo value ($\pm$ 1 standard deviation) on the July 13 survey was $0.48 \pm 0.04$, and albedo declined through each ~two-week period (Table 4). Glacier ice was exposed as the seasonal snowline moved upglacier in the following weeks, with albedo values decreasing to $0.16 \pm 0.11$ on August 22. The toe of the glacier is a high-accumulation area due to wind-blown snow deposition in the lee of a convexity (Adhikari and Marshall, 2013); it retained seasonal snow through mid-August, but was snow-free by August 22. On this final day of sampling, only the uppermost sampling site retained seasonal snow cover, possibly refreshed by a snow event on August 13-14.

The seasonal albedo decline is partly associated with the transition from snow to bare ice and partly because the glacier ice albedo systematically decreased over the course of the melt season, from an average value of $0.21 \pm 0.07$ on July 25 to $0.13 \pm 0.05$ on August 22 (Table 4). The snow albedo at the glacier toe also decreased from July 13 to August 9 (Figure 7b), but a decline in snow albedo was not apparent on the upper glacier. Much of the glacier surface had an albedo of ~0.1 by the end of the melt season, and the lower glacier had values of 0.07. As reported in previous studies (Brock et al., 2000; Klok and Oerlemans, 2002), ice albedo generally increases with altitude on Haig Glacier.

These ice albedo values are unusually low compared to values reported in the research literature and in the context of the longer-term record at Haig Glacier. There is an inheritance of accumulated particulate matter on the glacier surface from previous summers, but the significant changes from July 26 to August 22 indicate a strong intra-seasonal change, which feeds back on intensified melting and mass loss through the month of August. Some of this may be due to increasing concentration of impurities, as melting snow and ice leave the particulate load behind while the meltwater runs off.

In addition, we speculate that glacier darkening through the month of August may be associated with deposition of soot and other particulate matter associated with regional wildfires. The summer of 2017 was a severe wildfire season in western Canada. More than 1.2 million hectares of land burned in the province of British Columbia in 2017, a record at the time (Government of British Columbia, 2020). During our August field work on the glacier, the air smelled of smoke and visibility was limited due to smoky skies. Impurity measurements from snow and ice samples collected during the glacier visits indicate a ~four-fold increase in total carbon on the glacier surface from July 26 to August 9, 2017, from average concentrations of 5.6 to 22.7 mg $L^{-1}$, respectively (Table 5). The increase in impurities is evident in both inorganic and organic carbon. Particulate matter from local terrigenous dust also increased over this period, but by a factor of ~2.3 (Table 5). The mineral dust load is dominated by calcium and magnesium carbonate, with the carbon concentration associated with carbonaceous dust, $[C_{dust}]$, equal to 2.0 mg $L^{-1}$ on August 9. This indicates that more than 90% of the carbon on the glacier had a source other than local, terrigenous dust at this time, although we recognize that Ca and Mg are highly soluble and may have been preferentially removed by meltwater.

We are not able to partition the non-dust carbon between algal, wildfire, or other potential sources such as British Columbia industrial activity. The marked increase in impurity and total carbon concentrations from mid-July to mid-August, 2017 is consistent with the potential impacts of wildfire fallout on surface albedo. Wind direction data at the AWS confirms the prevalence of westerly winds bringing air masses from southern British Columbia to the study site. A full analysis of air mass trajectories and the specific source(s) of forest-fire fallout at Haig Glacier is beyond the current scope, but is recommended for followup studies.

## 4. Discussion

### 4.1 Albedo Modelling

Given the strong variation of surface albedo through the summer melt season – a typical decline from ~0.9 to ~0.2 from May to August – it is important to capture the seasonal albedo evolution in glacier hydrological and mass balance models. Numerous other researchers have tackled this problem, and physically-based snow albedo models have been proposed (e.g., Marshall and Oglesby, 1994; Flanner and Zender, 2006; Gardner and Sharp, 2010; Aoki et al., 2011).

In glacier modelling, the decrease in supraglacial snow albedo through the melt season can be approximated by a proxy for snow age or cumulative melting, to capture the systematic decline in albedo due to rounding and growth of snow grains, the effects of liquid water content in the snowpack, and increasing concentration of impurities (Brock et al., 2000). Following Hirose and Marshall (2013), we use a parameterization based on cumulative *PDD* to approximate the melt-season albedo decline on Haig Glacier,

$$\alpha_s = \alpha_0 \exp\left(-k \cdot PDD\right), \tag{7}$$

where $\alpha_0$ is the fresh-snow albedo (~0.85) and $k$ is an albedo decay coefficient. The free parameter, $k$, can be tuned to fit a given observational record such as those plotted in Figure 2b. The effects of snow grain size and impurity concentration can also be incorporated in $k$ in this type of model. Once the seasonal snow has melted, the albedo drops to that of the underlying firn or ice. Additional details can be added to the snow albedo model for thin snowpacks (e.g., less than ~10 cm), to capture the influence of the underlying firn or ice albedo when it begins to show through (Oerlemans and Knap, 1998).

This simple parameterization represents the seasonal albedo decline reasonably well, with minimal inputs, but does not capture the effects of fresh snow events, which temporarily increase the glacier surface albedo. The albedo impact of summer snowfall is ephemeral, but these events significantly increase the mean summer albedo and reduce the total summer melt, as discussed in Section 4.2. This is a difficult thing to model remotely or in future projections, as precipitation events can be extremely local in the mountains and the phase of precipitation (rain vs. snow) is difficult to predict. Rain and snow events are both are common on Haig Glacier in the summer months, often mixed on the glacier as a function of elevation.

As a simple approach to address this, we introduce summer snow events as a stochastic process. We randomly sample a normal distribution characterized by the mean and standard deviation of the expected number of summer precipitation events (Table 1), with the phase of precipitation determined by local temperature. We prescribe a linearly-decreasing snow fraction, $f_s$, between the temperatures of −2 and +2°C, for mean daily air temperature $T_a$:

$$\begin{aligned} f_s &= 1, & T_a &< -2°C \\ f_s &= 1 - (T_a + 2)/4, & T_a &\in [-2, 2°C] \\ f_s &= 0, & T_a &> 2°C \end{aligned} \tag{8}$$

Total event precipitation is also treated as a random variable. Summer snow events reset the surface albedo to the fresh-snow value, $\alpha_0$, and the albedo decay begins anew for the fresh summer snow, until it is ablated and the underlying surface re-emerges (old snow, firn, or ice).

The albedo of the underlying glacier firn or ice can be held constant or can be parameterized to decay with the amount of time exposed or as a function of impurity concentration. Lacking a good understanding or independent model of these processes, we assign the ice albedo to be

equal to the observed longterm mean at the Haig Glacier AWS site, 0.21 (Table 1). Temporal variability can also be introduced, based on random sampling of a normal distribution that describes the observed distribution of ice albedo values (Table 1).

Figure 8 plots an example of this simple treatment for summer 2007, with $k = 0.0009$ $(°C\,d)^{-1}$. The albedo model is embedded in a surface energy balance model (Ebrahimi and Marshall, 2016) that calculates *PDD* and melting, driven by the observed AWS data. The model is seeded with the observed winter snowpack, measured on April 13 of that year, and is run from May 1 to September 30. The timing of the transition from seasonal snow to ice is well-captured in Figure 8, and the number of fresh-snow events is in accord with the observations, but the timing is not correct (nor is it expected to be). Figures 8a and 8b show two different model realizations, illustrating the differences in timing of summer snow events.

The modelled summer snow events in Figure 8 are not completely random, as a stochastic precipitation event will only register as a snowfall when temperatures are cold enough. Because of this, snow events are more common in early and late summer and are also correlated in different model realizations. This temperature control also helps to capture the end of the summer melt season (beginning of winter snow accumulation) in the model. For instance, the end of summer, which occurs around September 17 in 2007, is well represented in Figure 8a but is ~one week late in Figure 8b. For the examples in Figure 8, the mean observed albedo values ($\pm$ $1\sigma$) in JJA and MJJAS are $0.53\pm0.23$ and $0.59\pm0.24$, compared with modelled values of $0.54\pm$ $0.24$ and $0.60\pm0.26$. Modelled albedo values are calculated from the mean of 10 model realizations; the plots in Figure 8 are representative of this population. Based on a simple t-test for comparison of means and Bartlett's test for equal variance (Snedecor and Cochran, 1989), the observed and model results are statistically equivalent.

**4.2 Implications for Glacier Mass Balance**

The seasonal albedo decline, summer snow events, and realistic values of firn and ice albedo are all important to resolve in models of glacier energy and mass balance. As an example, the energy balance model driven by the observed AWS meteorological data and surface albedo gives a total melt of 2.97 m w.e. for the 2007 melt season (May to September), the case study in Figure 8, corresponding to a mean melt-season surface albedo of 0.59. With the random snow events as illustrated in Figure 8, the mean of ten model realizations gives a summer melt of $2.82 \pm 0.25$ m w.e., corresponding to a mean melt-season albedo of 0.60. This slightly underestimates but is within the uncertainty of the observation-based estimate. Without the summer snow events, however, modelled melt equals 3.61 m w.e., with a mean melt-season albedo of 0.48. This overestimates summer melting and mass loss by 22%.

There are also important implications for modelling of glacier mass balance at remote sites or in future projections. As temperatures warm, summer precipitation events can be expected to shift to rain rather than snowfall, which would accelerate glacier melting. Without an explicit treatment of summer snow events and their impact on albedo, models calibrated to present-day

conditions will not capture this feedback. Similar caveats can be raised about assignment of a constant, observationally-based ice albedo in mass balance models; conditions vary between glaciers and may change in the future as a function of changing particulate loads, and possibly other factors. Physically-based models of impurity deposition and washout and the relation to ice albedo are needed for more reliable regional models and future projections. Similar efforts are underway to improve mass balance models for debris-covered glaciers (e.g., Reid and Brock, 2010; Rowan et al., 2015), although the processes differ for transport and dispersal of coarse debris such as rockfall.

Most studies to date involving regional or future models of glacier mass balance employ degree-day or temperature-index melt models, since in situ meteorological data are unavailable (e.g., Marzeion et al., 2014; Clarke et al., 2015). A full surface energy balance requires a large suite of variables such as wind speed, humidity, and cloud conditions. These are difficult to downscale from climate models with fidelity, relative to temperature fields. In this case, albedo does not appear directly in most formulations of the melt parameterization, but is implicit in the assignment of different degree-day melt factors for snow and ice, $d_{snow}$ and $d_{ice}$.

Observations of surface albedo evolution on mountain glaciers make it clear that a continuum approach is more appropriate, with changing degree-day melt factors that track the seasonal albedo evolution (e.g., Arendt and Sharp, 1999). The monthly melt factor $d$ was calculated from Eq. (6) using mean monthly values of melt energy and temperature at the Haig Glacier AWS from 2002-2015, giving a range of values that can be compared directly with mean monthly albedo (Figure 9). As expected, there is a relatively strong inverse relation, with a linear correlation coefficient of $-0.66$ and a coefficient of determination of $R^2 = 0.44$. A linear fit to the data gives the regression equation $d = 7.98 - 6.16\alpha_s$, significant at $p < 0.0001$. This relation could be applied if one has independent estimates of the surface albedo, e.g., from remote sensing (e.g., Williamson et al., 2016) or drone surveys. Alternatively, Table 2 includes mean monthly values of the melt factor for the full dataset, which would be preferable to using single values for snow and for ice. Monthly factors could also be calculated for the radiation melt coefficient in enhanced temperature-index melt models which use potential direct solar radiation as an input (e.g., Hock, 1999; Clarke et al., 2015; Carenzo et al., 2016).

The correlation matrix in Table 3 summarizes the broader relations between albedo, temperature, and mass balance conditions on Haig Glacier. The monthly net energy and melt are strongly correlated with temperature and $PDD$ ($r \sim 0.9$), implying that temperature-index melt models could give good estimates of monthly melt at this site, given a judicious choice of melt factor, $d$. The relation between $Q_N$ and melt is weaker but still significant for the total summer melt ($r \sim 0.7$). Annual mass balance is highly correlated the summer balance ($r = 0.94$), emphasizing the importance of the summer melt season, which in turn is highly sensitive to surface albedo. Mean summer albedo is highly correlated with net annual mass balance ($r = 0.76$). This is stronger than the correlation of net balance with mean summer temperature or $PDD$ totals. Closely related to this, summer snow events have a significant association with the summer and net mass balance ($r = -0.73$ and $r = 0.70$, respectively). The amount of mass added to the glacier is small in summer

relative to the winter snowpack (less than 5%), but net mass balance is more strongly correlated with the number of summer snow events than the winter balance, due to the albedo impact.

**4.3 Temporal Variability and Trends in Ice Albedo**

Glacier ice albedo is low at this site, in association with high concentrations of supraglacial impurities. The impurities are a combination of mineral dust, primarily calcium and magnesium carbonate, other sources of inorganic carbon, and organic carbon, including active algal populations. The mean value of summer ice albedo at the AWS site is 0.21, but this dips to 0.11 in some years (2003, 2015, 2017), possibly in association with regional wildfire activity. The summers of 2003 and 2017 were particularly active wildfire seasons in southern British Columbia, upwind of our study site, and ice albedo declined through the summer melt season in these two summers. This was unusual, however; overall, there is no evidence of decreases in ice albedo over the melt season (Figure 6a). In contrast, bare-ice albedo increases slightly in the late summer and early autumn, perhaps in association with superimposed ice formation and/or meltwater rinsing of the glacier surface after transient summer snow events.

Similarly, there is no multi-year trend for ice albedo at this site (Figure 6b), although this record is limited to a relatively short period (2002-2015) at just one location on the glacier. Based on the available data, however, there is no evidence of glacier 'darkening' over the period of study, despite years such as 2003 which experienced heavy deposition and accumulation of particulate matter. This stands in contrast to reported glacier albedo reductions over the last two decades in other regions (e.g., Mernild et al., 2015; Naegeli et al., 2019). These results imply that there is some degree of effective cleansing and refreshing of the glacier surface through rainfall and meltwater runoff, although the baseline albedo remains low and there is a multiyear accumulation of supraglacial impurities over much of the glacier.

The transect data from August 22, 2017 indicate lower albedo values and greater impurity loads near the glacier terminus (Figure 7b; Miller and Marshall, in preparation). This is consistent with increased concentration of residual particulate matter due to cumulative melting, within a given summer or over many years, as well as the possibility of greater mineral dust loading on the lower glacier (and associated nutrients to support algal activity), as reported by Oerlemans et al. (2009). We do not have the data to assess multiyear albedo or impurity trends on the lower glacier, to test whether the terminus zone is darkening as a feedback to negative mass balance trends, as has been reported elsewhere (Oerlemans et al., 2009; Naegeli et al., 2019).

At a given location on the glacier, increases in the concentration of impurities during the melt season exceed what would be expected from melting. As an example, total carbon concentrations across the glacier increased four-fold from July 26 to August 9, 2017 while mineral dust concentrations more than doubled (Table 5). Applying the surface energy balance at an elevation of 2730 m on the upper glacier, we estimate a total melt of 0.48 m w.e. over this two-week period. This is not enough to explain the observed increases in concentration, as subsurface snow samples had carbon and dissolved ion concentrations below the detection limit of 0.1 mg L$^{-1}$.

Meltwater runoff should also lead to leaching and removal of some dissolved ions. The increased particulate matter must have been due to deposition on the glacier. Further work is needed to quantify deposition and rinsing (leaching, washout) of particulates, to develop models for these processes, and to characterize their influence on temporal and spatial variations in albedo.

## 5. Conclusions

Albedo measurements from the upper ablation area of Haig Glacier over the period 2002-2017 indicate significant interannual variability in mean melt-season and glacier ice albedo, but there is no temporal trend in surface albedo over this period. This runs counter to documented albedo reductions elsewhere (Oerlemans et al., 2009; Mernild et al., 2015; Williamson et al., 2019; di Mauro et al., 2020), and to anecdotal evidence of darkening glaciers in the Canadian Rockies. The result may just be specific to the Haig Glacier AWS site; we do not have data to constrain albedo trends in the lower ablation zone, where the ice albedo is lowest and glacier thinning has been most extensive over the observation period. The observational record is also short for trend detection. Haig Glacier mass balance has been negative through this whole period, and it is possible that the glacier has darkened over a multi-decadal time frame (e.g., since the 1970s), but not since the early 2000s.

The baseline summer ice albedo at the Haig Glacier AWS site is $0.21 \pm 0.06$, so it has been relatively low for the whole period of study. It drops to values as low as 0.11 in some years (2003, 2017), in association with strong wildfire seasons in southern British Columbia, upwind of our study site. Large increases in total and organic carbon concentrations measured on the glacier in August 2017 support this association. The glacier ice appears to recover from these low-albedo summers, however, returning to albedo values of ~0.2 in subsequent years. This is evidence of cleansing of the glacier surface by rainfall and meltwater runoff. Impurities at Haig Glacier are dominated by fine particulate matter, mineral dust in particular and much of this may be effectively leached as dissolved sediment load. The mass balance of supraglacial particulate matter is not well understood, and requires further study.

Other processes controlling the variability in ice albedo also require further study. We find no relation between mean daily ice albedo and cloud conditions in our data, as reported elsewhere (e.g., Brock, 2004; Abermann et al., 2014) and as theoretically expected. This may be because the albedo is relatively low, with a high concentration of impurities; particulate matter and liquid water content act as isotropic absorbers, reducing the sensitivity of specular reflection to zenith angle (hence, diffuse vs. direct radiation). We also see no evidence of ice albedo reductions through the melt season, unlike in seasonal snow, although the major wildfire years provide an exception to this. This further argues for effective rinsing of the glacier surface in most summers, save when dry deposition of particulate matter is unusually high.

We do see evidence of temporary increases in bare-ice albedo to values of ~0.3 following melting and runoff of fresh summer snow. The post-snowfall glacier ice albedo is commonly about 0.15 higher than before the snow event. This may be due to a reflective, superimposed ice

crust that temporarily forms after snow events, or it could be a result of effective washing of the glacier surface from the melting of clean snow. The increase in ice albedo is transient, but the effect persists for two or more days after the new snow has melted away.

Overall, summer snow events at Haig Glacier have a large impact on mean summer albedo and glacier mass balance. An average of $9.3 \pm 2.6$ such events were recorded each summer, resulting in a mean melt-season albedo increase of 0.12 (e.g., from 0.48 to 0.60 in 2007). Such events are particularly significant when they occur late in the summer, temporarily brightening the low-albedo ice. Based on both energy balance modelling and direct AWS observations, we estimate that summer snow events reduce summer melting and mass loss by about 20% at Haig Glacier. This is an important potential feedback and sensitivity to climate change, as warming is likely to cause more of these summer precipitation events to shift to rainfall rather than snow in the coming decades.

We introduce a stochastic model of summer snow events in a simple model that captures the typical melt-season albedo evolution on glaciers. This is necessary for realistic mass balance modelling at Haig Glacier, and could be adapted for use elsewhere, as long as there is some knowledge of precipitation frequency during the melt season. The seasonal albedo evolution on glaciers governs how effectively incoming shortwave radiation is converted to melt, and it is important to capture this influence in simplified melt and mass balance modelling applications where local meteorological data are not available. For temperature-index melt modelling, we suggest that either monthly melt factors or melt-factor parameterizations as a function of albedo can better capture the conversion of positive degree days to melt.

Haig Glacier albedo values and summer snow conditions may not be broadly applicable, particularly given regional differences in the provenance and concentration of impurities. Particulate loading is highly variable in space, even within a given glacier. The processes discussed in this contribution and the general pattern of melt-season albedo evolution are relevant to most mountain glaciers, however. These observations can help to inform regional models of glacier mass balance and assessments of glacier response to climate change. We emphasize the need for process studies of particulate mass balance (deposition, accumulation, transport, and removal) in supraglacial environments, including the potential effects of forest fire fallout on glacier albedo and mass balance.

**Acknowledgements**

We are grateful to the Natural Science and Engineering Research Council of Canada (NSERC) and the Canada Research Chairs program for sustained, long-term support of the Haig Glacier project. Rick Smith of the University of Calgary Weather Research Station has been instrumental in helping to maintain and calibrate our sensors. Numerous graduate and undergraduate students assisted with the Haig Glacier fieldwork since 2000, and we particularly thank Patrick Coulas for his assistance with the summer 2017 albedo and supraglacial snow/ice sampling.

**Code/Data Availability**

Mean daily values of the automatic weather station data from Haig Glacier are published in the University of Calgary data repository (Marshall, 2020), doi:10.5683/SP2/7CXPPI. MATLAB code for the energy balance model used in this study is available from the author on request.

**Author Contributions**

SM initiated the Haig Glacier field study, led the field effort and data collection, wrote the MATLAB code for the surface energy balance modelling, was responsible for the data analysis and wrote the manuscript. KM collected the summer 2017 surface albedo and supraglacial chemistry data as part of her Masters research at the University of Calgary.

The authors declare no competing interests and no conflict of interest with this research or its conclusions.

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

## Tables

**Table 1**. Mean summer albedo and mass balance conditions at Haig Glacier, 2002-2017, based on glacier-wide simulations driven by local AWS data. $B_w$, $B_s$, and $B_n$ are the winter, summer and annual specific mass balance, $N_m$ and $N_{ss}$ are the number of melt days and summer snowfall days from May through September, $PDD$ are the positive degree days over the summer melt season, $T$ and $Q_N$ are the mean June through August (JJA) air temperature and net energy flux, and $E_m$ is the total summer melt energy. Albedo values are measured at the glacier AWS: $\alpha_s$ is the mean JJA surface albedo and $\alpha_i$ is the measured ice albedo for the composite of snow-free days ($N = 224$).

| | $B_w$ (m w.e.) | $B_s$ (m w.e.) | $B_n$ (m w.e.) | $N_m$ | $N_{ss}$ | $PDD$ (°C d) | $T$ (°C) | $Q_N$ (W m$^{-2}$) | $E_m$ (GJ m$^{-2}$) | $\alpha_s$ | $\alpha_i$ |
|---|---|---|---|---|---|---|---|---|---|---|---|
| mean | 1.35 | −2.60 | −1.25 | 137 | 9.3 | 671 | 5.3 | 107 | 1.113 | 0.55 | 0.21 |
| std dev | 0.24 | 0.62 | 0.68 | 8 | 2.6 | 92 | 0.8 | 17 | 0.177 | 0.07 | 0.06 |

**Table 2**. Mean monthly temperature, surface energy balance, and melt conditions at the Haig Glacier AWS, 2002-2015. Symbols are as in Table 1, with the addition of degree-day melt factors, $f_E$, calculated from Eq. (7), and the conventional version, $f_{PDD}$, calculated from Eq. (6).

| Month | $T$ (°C) | $PDD$ (°C d) | $Q_N$ (W m$^{-2}$) | $E_m$ (MJ m$^{-2}$) | $\alpha_s$ | melt (m w.e.) | $f_E$ | $f_{PDD}$ (m w.e. (°C d)$^{-1}$) |
|---|---|---|---|---|---|---|---|---|
| May | −1.0 | 42 | 22 | 47 | 0.77 | 0.13 | 3.4 | 3.3 |
| June | 2.8 | 100 | 62 | 142 | 0.71 | 0.40 | 4.4 | 4.2 |
| July | 6.8 | 212 | 126 | 319 | 0.56 | 0.93 | 4.5 | 4.4 |
| August | 6.1 | 191 | 137 | 368 | 0.38 | 1.10 | 5.8 | 5.8 |
| Sept | 2.0 | 91 | 42 | 116 | 0.64 | 0.35 | 3.7 | 3.7 |

Table 3. Linear correlation coefficients for albedo, energy balance, and mass balance conditions at Haig Glacier, 2002-2015. Symbols are defined in Tables 1 and 2. The top right sector is for mean summer (JJA) and annual mass balance conditions ($N = 14$), and the bottom left sector (italicized) shows correlation coefficients for all available monthly mean values from May to September ($N = 71$). For the monthly data, $B_s$ refers to the monthly summer balance (melting minus refreezing), defined as a negative for mass loss. $B_n$ and $B_w$ are not relevant for the monthly data. Correlations that are not significant at the 95% level [$p > 0.05$] are shown in brackets.

| | Annual ($B_n$, $B_w$) or summer (JJA, all other variables) means or totals | | | | | | | | |
|---|---|---|---|---|---|---|---|---|---|
| | $\alpha_s$ | $B_w$ | $B_s$ | $B_n$ | $N_{ss}$ | PDD | T | $Q_N$ | $E_m$ |
| $\alpha_s$ | — | [0.39] | 0.74 | 0.86 | 0.67 | [−0.30] | −0.68 | −0.81 | −0.67 |
| $B_w$ | | — | [−0.04] | [0.47] | [0.20] | [0.18] | [0.12] | [−0.12] | [0.01] |
| $B_s$ | *0.89* | | — | 0.93 | 0.73 | −0.68 | −0.75 | −0.93 | −0.98 |
| $B_n$ | | | | — | 0.70 | [−0.43] | −0.64 | −0.87 | −0.82 |
| $N_{ss}$ | | | | | — | −0.58 | −0.58 | −0.76 | −0.73 |
| PDD | *−0.74* | | *−0.89* | | | — | 0.63 | 0.65 | 0.76 |
| T | *−0.73* | | *−0.88* | | | *0.97* | — | 0.86 | 0.73 |
| $Q_N$ | *−0.84* | | *−0.97* | | | *0.92* | *0.91* | — | 0.92 |
| $E_m$ | *−0.87* | | *−0.99* | | | *0.92* | *0.90* | *0.99* | — |
| $d_E$ | *−0.66* | | *−0.62* | | | *0.39* | *0.41* | *0.64* | *0.65* |

Table 4. Mean albedo values (±1σ) along the Haig centreline transect during four surveys in summer, 2017: glacier average, and for the subset of sites over seasonal snow and glacier ice. The number in brackets indicates the number of samples for each average. Snow values on the upper glacier are estimated on August 9.

| Date | All sites | Snow | Ice |
|---|---|---|---|
| July 13 | 0.48 ± 0.04 (33) | 0.48 ± 0.04 (33) | — |
| July 26 | 0.34 ± 0.15 (33) | 0.48 ± 0.05 (16) | 0.21 ± 0.07 (17) |
| August 9 | *0.23 ± 0.11* (33) | *0.41 ± 0.03* (9) | 0.17 ± 0.05 (24) |
| August 22 | 0.16 ± 0.11 (33) | 0.47 ± 0.08 (3) | 0.13 ± 0.05 (30) |

**Table 5**. Mean concentrations ($\pm 1\sigma$) of major ions and carbon along the Haig centreline transect surveys on July 26 and August 9, 2017 ($N = 11$). All concentrations have units mg L$^{-1}$. TC is total carbon, TIC and OC are inorganic and organic carbon, [C$_{dust}$] is the inorganic carbon associated with carbonate mineral dust, and [dust] is the total mineral dust. Factor indicates the ratio of concentrations for August 9 over July 26.

| Date | [Ca$^{+2}$] | [Mg$^{+2}$] | [TC] | [TOC] | [IC] | [C$_{dust}$] | [dust] |
|------|-----------|-----------|------|-------|------|-----------|--------|
| July 26 | $2.3 \pm 2.9$ | $0.3 \pm 0.2$ | $5.6 \pm 5.7$ | $3.7 \pm 3.4$ | $1.9 \pm 2.3$ | $0.9 \pm 1.0$ | $7.1 \pm 7.8$ |
| August 9 | $5.1 \pm 6.9$ | $0.9 \pm 1.5$ | $22.7 \pm 13.6$ | $11.3 \pm 8.4$ | $11.4 \pm 6.3$ | $2.0 \pm 2.7$ | $16.4 \pm 21.4$ |
| Factor | 2.2 | 2.7 | 4.0 | 3.0 | 6.0 | 2.3 | 2.3 |

    **Figures**

6    **Figure 1.** (a) Haig Glacier study area in southwestern Canada (inset b). (b) Haig Glacier is about
7    100 km southwest of Calgary, AB, on the eastern slopes of the Canadian Rocky Mountains. (c)
8    Map view of the glacier, indicating the locations of the two automatic weather stations (AWSs).
9    All images are courtesy of Google Earth ©.

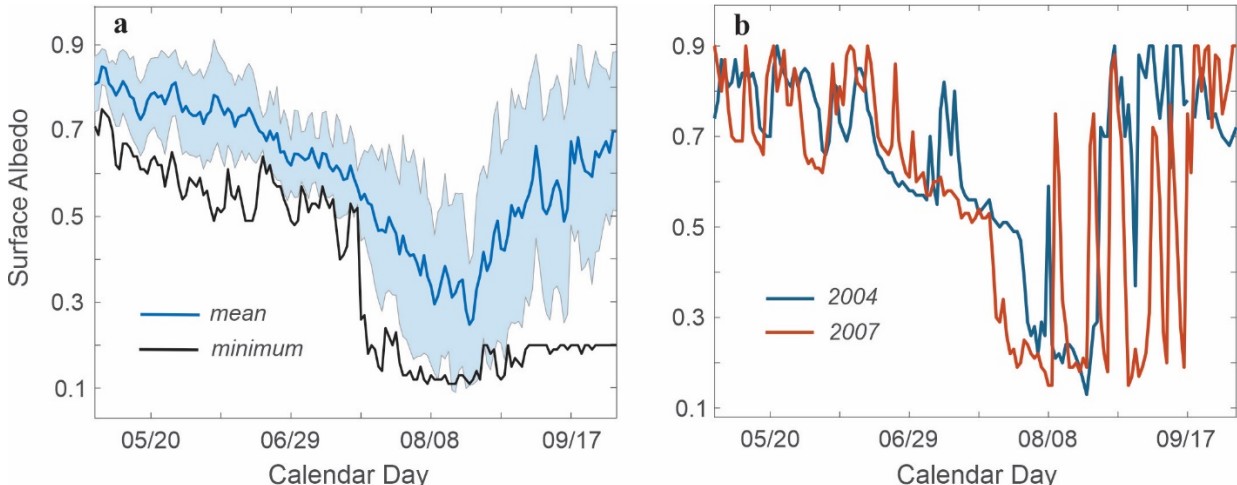

**Figure 2.** Daily albedo evolution at the Haig Glacier AWS site over the melt season, May to September. (a) Mean and minimum daily albedo from 2002-2016. Shaded area indicates the 1-standard deviation range about the mean. (b) Select individual years (2004 and 2007) to better illustrate the transition from seasonal snow to exposed glacier ice and the albedo spikes associated with summer snow events.

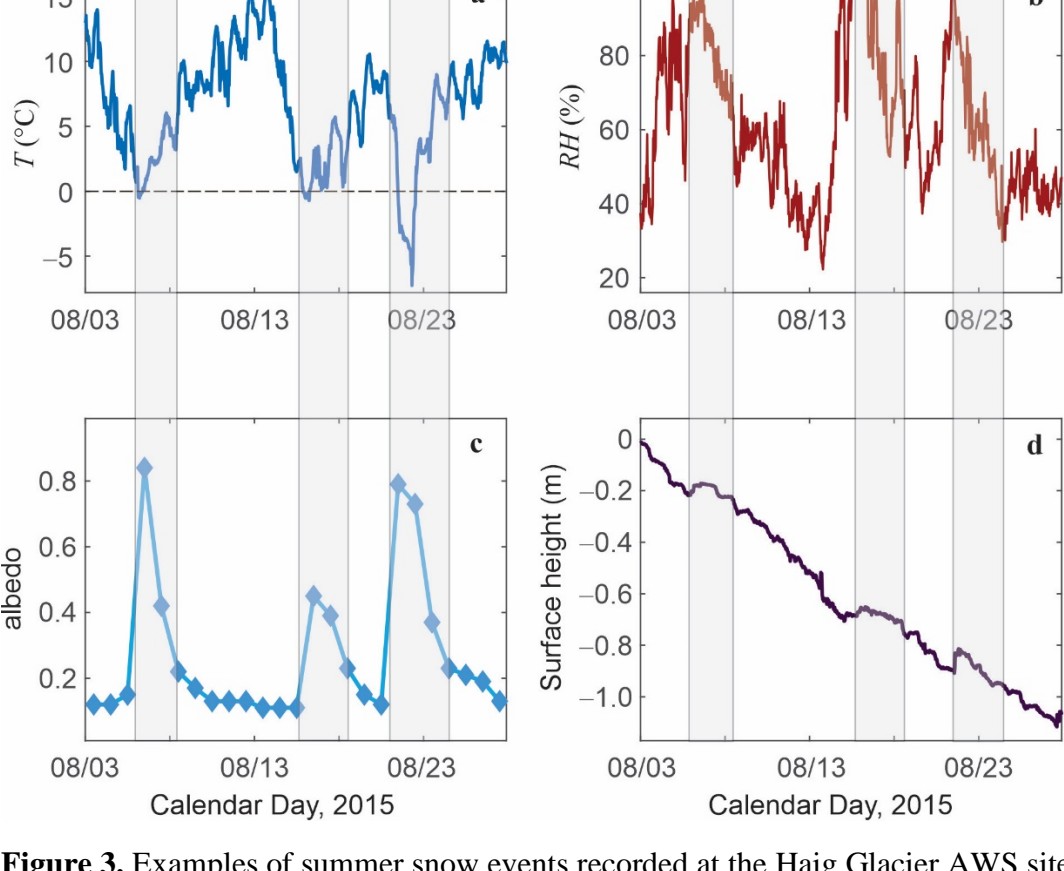

**Figure 3.** Examples of summer snow events recorded at the Haig Glacier AWS site from August 3-28, 2015. (a) Air temperature, (b) relative humidity, (c) mean daily albedo, and (d) surface height, as measured by the ultrasonic depth gauge (SR50).

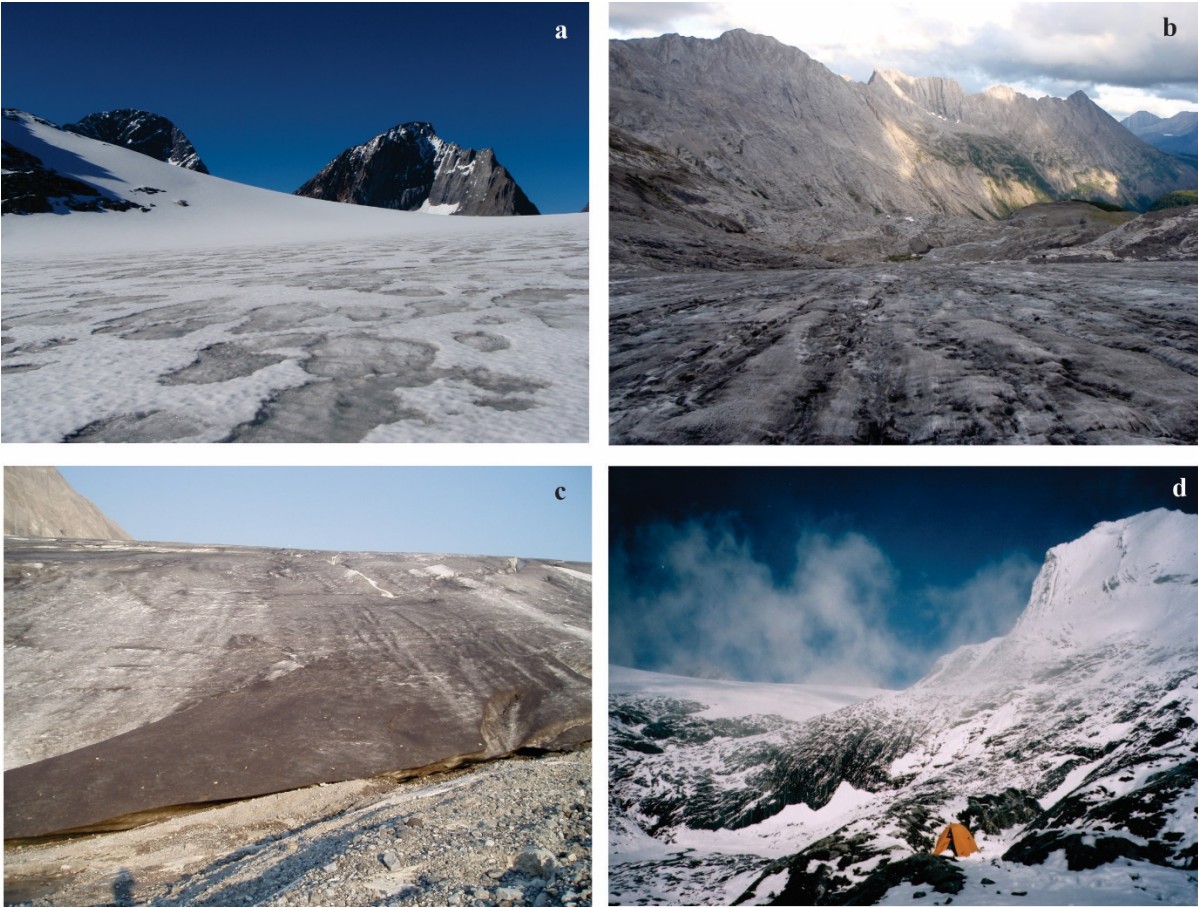

**Figure 4.** Photographs of the Haig Glacier, illustrating the variability of summer surface cover. (a) The transition from seasonal snow to exposed glacier ice. (b) Meltwater runnels looking downslope in the ablation area, illustrating the heterogeneous but extensive concentration of surface impurities. (c) Dark ice at the glacier terminus. (d) Fresh snow covering the glacier after a heavy August snowfall. Photos by S. Marshall.

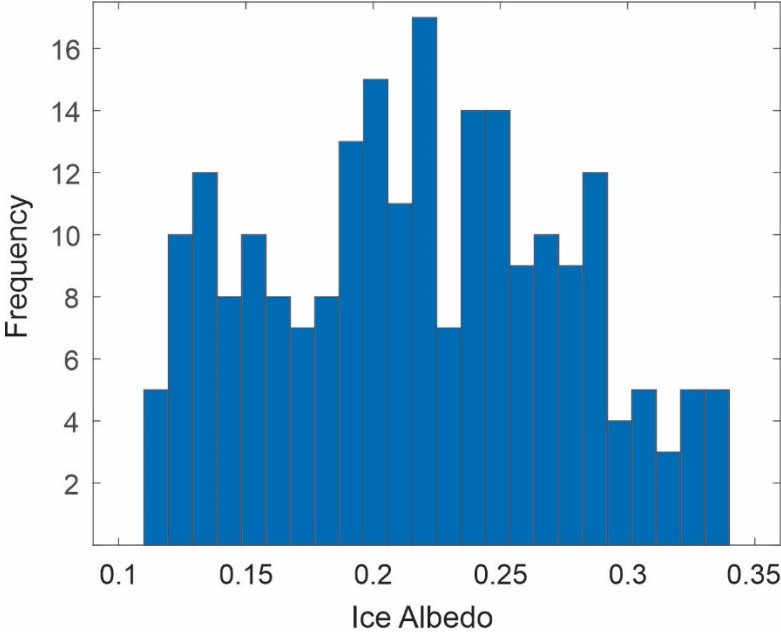

**Figure 5.** Distribution of daily mean ice albedo values recorded at the Haig Glacier AWS from the summers (JJA) of 2002 to 2015, for all days that were snow-free at the AWS site (*N* = 224).

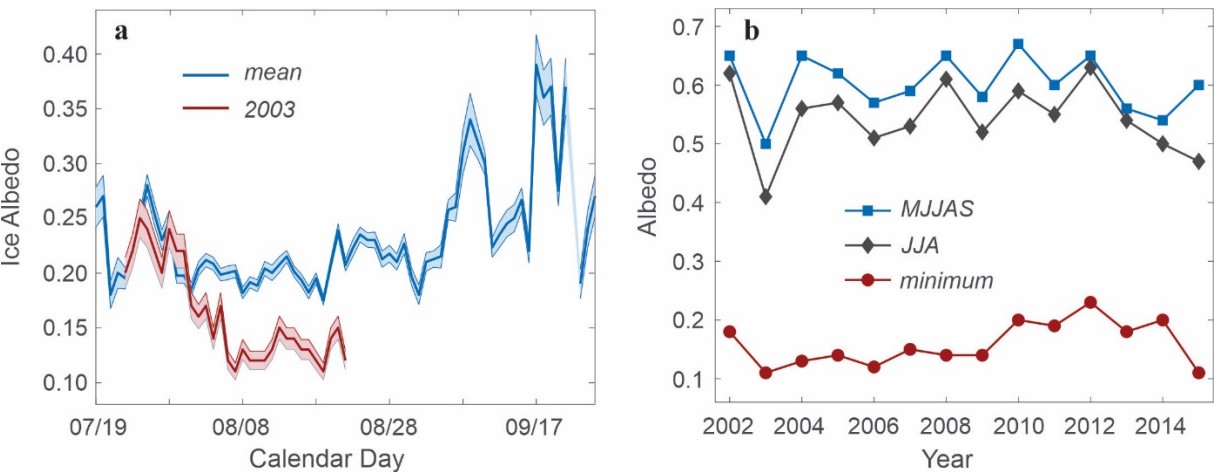

**Figure 6**. (a) Evolution of Haig Glacier ice albedo through the summer melt season. Mean values for 2002-2015 are shown in blue and available data from summer 2003 is in red. In 2003, the station was leaning too much for reliable data after August 23. Shading indicates the uncertainty envelope of the measurements (one standard error). (b) Evolution of the mean melt-season albedo at Haig Glacier from 2002 to 2015, for May through September, July through August, and the minimum daily value of each year.

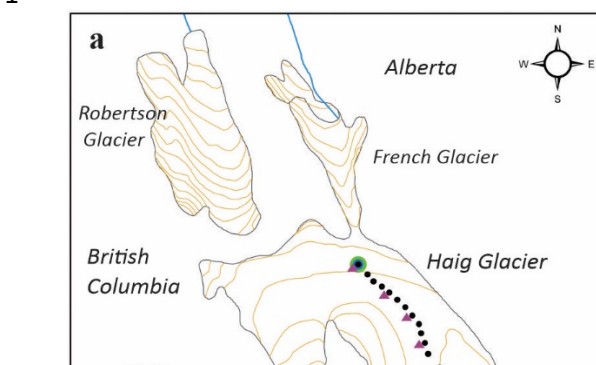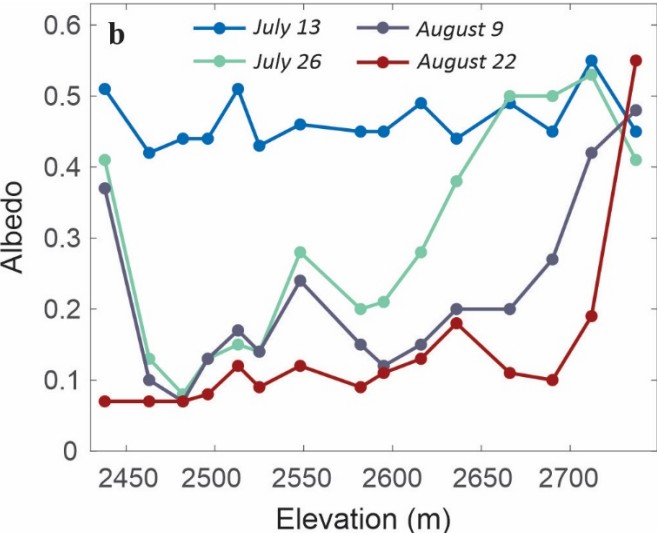

**Figure 7**. (a) Black circles show the 33 survey sites for winter mass balance and albedo measurements along the Haig Glacier 'centreline' transect. (b) Evolution of surface albedo along the centreline transect during four visits in July and August, 2017.

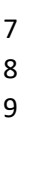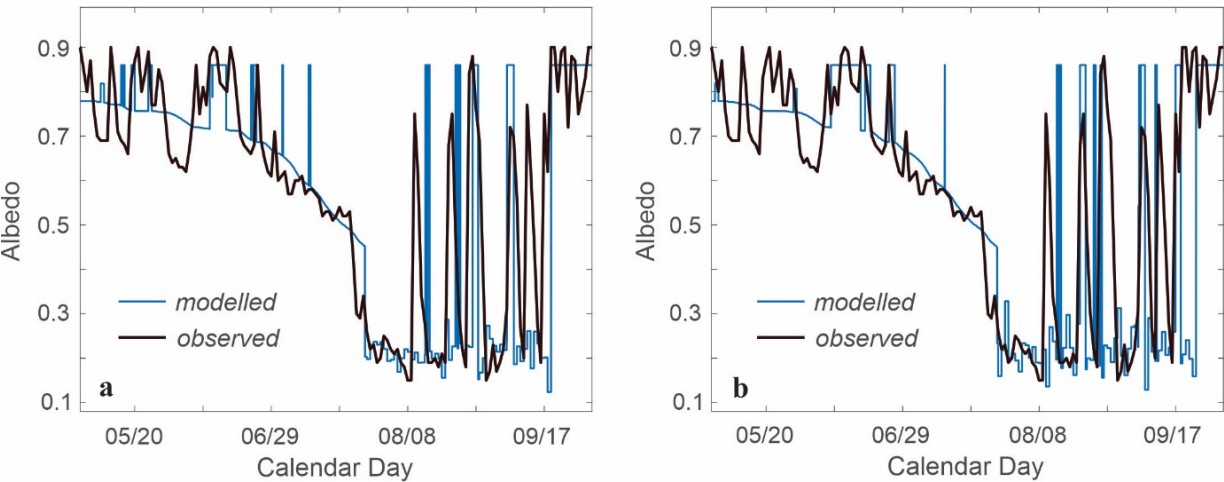

**Figure 8**. Two realizations of modelled vs. observed surface albedo at the Haig Glacier AWS site, May 1 to September 30, 2007. Summer snow events (albedo spikes) are modelled as random events in the albedo model.

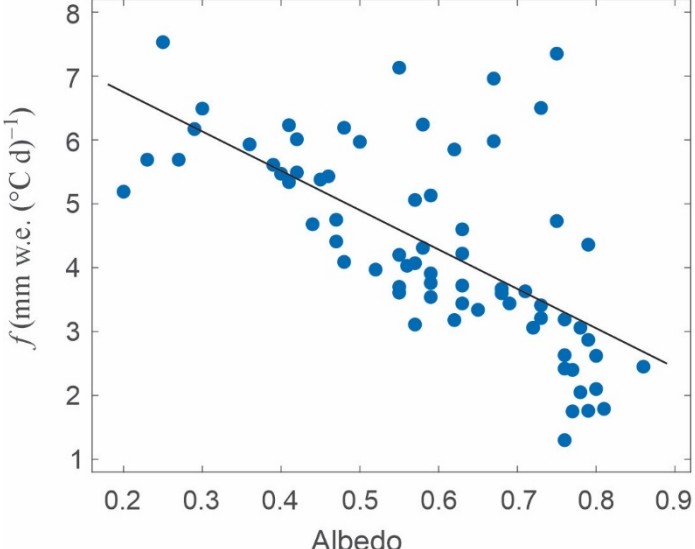

2
3  **Figure 9**. Degree-day melt factor, *f*, as a function of monthly mean albedo and melt energy at the
4  Haig Glacier AWS site, for MJJAS from 2002 to 2015.