# Peer review of "Seasonal and Interannual Variability of Melt-Season Albedo at Haig Glacier, Canadian"

_The Cryosphere, 2020_

## Referee Comment (RC1) · Jing Ming (Referee) · 7 May 2020

Jing Ming Beacon Science & Consulting Melbourne, Australia Email address petermingjing@hotmail.com

The paper uses a long-term observation dataset of surface albedo in the Haig Glacier during the period 2002-2017 to depict the seasonal and Interannual Variability of Melt-Season Albedo at Haig Glacier, Canadian Rocky Mountains. It is important to present this valuable dataset for developing any energy or mass balance model to project the evolution of the glacier. The tuning of the MB model is also a nice try. The paper is promising to be finally accepted by the Cryosphere from my point of view. However,

before its formal acceptance, I want to address a few concerns here.

Specific comments: 1. The first paragraph of the Introduction part seems to describe the target of the work, which is more proper to be moved to the end of this part. 2. Line 3-4. The sentence, "Variations in surface albedo, therefore, exert a strong control on the surface energy balance and available melt energy", needs a reference. Here is one by Ming et al. (2015) for your information. - Ming, J., et al. (2015). Widespread albedo decreasing and induced melting of Himalayan snow and ice in the early 21st century. PLoS One. 10: e0126235. 3. Line 4. "manuscript" -> "work" or "study". 4. Line 44-51. This paragraph reads to be wordy and not well organized and needs to be rephrased. 5. Line 44. The word "this" is not clear. Please clarify it. 6. Line 45-51. These two sentences are too long to read. Please rephrase them to several shorter sentences. 7. Line 97. Figure 1 had better incorporate a smaller map of the study area from a global perspective so that the readers could know where the study area is in the first sight. It is also beneficial to include the conditions of climatology for this area in the figure. 8. Line 97. "Albeta" -> "the Albeta province" or "the Albeta state" or "the Albeta city"? 9. Line 101-102. "Snow surveys conducted on the glacier each May indicate a mean winter snowpack of 1.35 m water equivalent (w.e.) on the glacier from 2002-2017, with a standard deviation ($\sigma$) of 0.24 m w.e. (Table 1)." Is this original from this study or cited from other studies? If it is in the latter case, it needs a reference. I suggest using a simpler expression of 1.35 $\pm$ 0.24 m w.e. to replace the long one in the previous form. 10. Line 105. Could you also add a standard error of the mean of the temperature after the number 5.3 âĐČ? 11. Line 111-115. This paragraph could be incorporated into the measurement section, and the next as well, because two paragraphs are more like introducing the measurement and data collection. 12. Line 116. "The forefield AWS" -> "The AWS in the forefield"? This phrase appears a few times throughout the text. 13. Line 123. Please clarify what "the set of available in situ data" is. 14. Line 133-134. Here needs a more detailed description of how to do manual quality control and remove the questionable data, although the authors claimed that the data control had been introduced in Marshall (2014). The current explanation

is too simple to understand the method. 15. Line 135. "concentrates" -> "focuses" or "zooms in"? The usage of "concentrate" here seems to be strange. 16. Line 135-136. The intent of the sentence is unclear, and please rephrase it. 17. Line 136. "pragmatic" -> "virtual"? 18. Line 137. "evolution" -> "variation"? 19. Line 142. "than" -> "from" or put "other" before it. 20. Line 150. Please clarify how you calculated out 7%. 21. Line 157. The last sentence "modelling of potential reflected radiation from valleys walls indicates that this is negligible at our AWS site". Could you please present evidence of your claim? 22. Line 159. "paper" -> "work". "repeat" -> "repetitive" or "repeated". 23. Line 161. "Haig Glacier albedo" -> "The albedo of the Haig Glacier". 24. Line 162. "points" (geometric concept) -> "sites" (geographic concept). Check that throughout the context. 25. Line 166. Was the sensor held manually? If so, how did you avoid the shadow of the body when measuring? Please clarify. 26. Line 167. Please give the detail of presuming a 10% uncertainty. 27. Line 169. "for melting and major ion and organic carbon analyses" -> "for the analysis of major ions and organic carbon". Please provide the source or references of the impurities used in this work. 28. Line 176. "data" -> "temperature" and "precipitation"? Please specify them. 29. Line 177. Please check the use of articles throughout the context. "forefield AWS data" -> "the data from the AWS in the forefield". 30. Line 191. What do you mean "the net energy goes to melting"? Please rephrase it. 31. Line 195. Give out the exact value of Lf (334 J g$-1$). 32. Line 240. Please clarify the definitions of a and b, respectively. 33. Line 430. The first sentence needs to be rephrased. Do you mean "the impact of fresh snow on albedo"? 34. Line 450. "forced" –> "driven". 35. Table 1. Please clarify the definitions of summer and winter for this study in the caption or context. 36. Figure 2. Why didn't the authors use the lines of means with shaded area indicating the error? 37. Figure 7. The blue points denoting the snowpits are blur. 38. Figure 8. What about the significances between the observed and modelled? 39. Figure 9. The same issue as that in Figure 8. Significance? 40. The language of the context needs a thorough check for grammar and misused words, such as articles, the function word "of", ambiguous statements, etc.

---

## Referee Comment (RC2) · Anonymous Referee #2 · 16 Jun 2020

Review of: Seasonal and Interannual Variability of Melt-Season Albedo at Haig Glacier, Canadian Rocky Mountains

Submitted to The Crysphere by Marshall and Miller.

Major Revisions required.

Albedo measurements from in-situ weather stations are used to identify melt season albedo dynamics for Haig Glacier. The results are used to comment on the conventional application of degree-day melt rates and on how albedo describes glacier mass balance.

[Figure]

These types of in-situ data driven papers are very important to the understanding of glacier dynamics and glacier mass balance, especially for mountain glaciers. The manuscript is well written with a logical presentation of material. I would suggest a minor re-organisation of the Introduction section to separate the literature review from specific mention of the study on Haig Glacier, as the sporadic reference to the study on Haig Glacier comes across as a bit disjointed. The final paragraph of the Introduction should be devoted to specific details regarding Haig Glacier. Specifically, how the study on Haig Glacier addresses the limitations related to glacier albedo and modelling.

Abstract:

The improvements related to the stochastic model on mass balance and the modification of the degree day model should be provided in more detail.

Line 11: Summer should be defined in the abstract (e.g., June 1 to August 31). Summer is defined on Line 104.

Body of text:

Line 28: It is true that albedo is involved in the control of surface energy balance, but it is the net radiation (short wave and long wave) that mostly controls melt. Net radiation was previously mentioned, but a better description of how net radiation is related to albedo and what the proportion of shortwave to longwave radiation is, would be very useful.

Line 29-30: Reference to Haig Glacier should probably come at the end of the introduction.

Line 48-51: This sentence seems to be a bit misplaced and should be moved to the end of the introduction as a bridge between the literature review and the methods section.

Line 64: Please define what a melt-albedo feedback is.

Line 69-71: Snow algae can be of many species (up to 4 or 5). Is there a reference for

this material, or is it an observation from Haig Glacier? If it is an observation it would find a better home in the Results section.

Line 76: A recent article in Remote Sensing of Environment might be of interest here: Williamson et al., 2020 - Comparing simple albedo scaling methods for estimating Arctic glacier mass balance.

Line 78-79: This material might be better suited in the final paragraph of the Introduction.

Line 82: Can you provide more detail on what the "simplified parameterizations" entails?

Line 111: Campbell does not make many instruments. The details for the instruments should be included (manufacturer and instrument), at the very least for the radiometric instruments, as different instruments are sensitive to different range of the EM spectrum.

Line 119-120: Data collection ongoing has previously been mentioned.

Line 124: If only one station is collecting data how was the lapse rate estimated? Please provide details.

Line 126: How much error is related to the estimation?

Line 132: Define "questionable data".

Line 154: There is a recursive reflection from the bottom of optically thin clouds or from scattered clouds and a high albedo snow covered surface.

Line 165: Please define Jaycar QM1582. What is the spectral range of this instrument?

Line 221: "this" should be these.

Line 245: The introduction mentions two AWS. It is not clear which station these results refer to. I assume from the data period this is the on ice station (upper ablation zone).

Line 284: "jump" might not be the best descriptor here.

In Table 2 why is Em larger for August than July? Cloud cover – because Em is using only shortwave radiation?

Line 289: What type of regressions are these? Linear, least-square regressions, Pearson's? Are the correlations statistically significant? If so, which ones?

Line 291: What does "correlated" mean in this instance?

Line 297: Define "fewer samples".

Line 300: Define "melt out" or replace with better descriptor.

Line 309: Define "ripened and saturated"? Line 315: Some other citations that might be useful here, especially in the context of spatial variability of albedo. 1. B.W. Brock, I.C. Willis, M.J. Sharp. Measurements and parameterization of albedo variations at Haut Glacier d'Arolla Switz. J. Glaciol., 46 (2000), pp. 675-688 2. S.N. Williamson, L. Copland, D.S. Hik. The accuracy of satellite-derived albedo for northern alpine and glaciated land covers Polar Sci., 10 (2016), pp. 262-269

Line 323: Describe the film, thickness composition, etc. Is there liquid water in the surface matrix? If so, what effect does this have on albedo? O.k., I see this is addressed on Line 335. Line 325: Not clear where the values for Figure 5 are coming from, and provide how N=224 was derived. Line 343: Adding year to the dates will reduce confusion. Figure 6: Mean values should have standard error included on the figure. Line 376: "dropping" should be decreasing. Line 385: The values of ∼0.1 and 0.07 are close enough that instrument error might render these inseparable? Line 395: Can evidence be presented that Haig glacier was indeed downwind of the forest fire smoke? For example, can specific fire events be linked to specific albedo declines for 2017? Without this link the material presented here is speculation. Line 399: Which year? Line 400: Please present pertinent details for the data. The reader can't evaluate the data from an unpublished source. Line 401: How is this "consistent"? Provide

details, references or rationale. Line 407: I assume that algae assimilate carbon that was on the glacier before, or during, its growth. If this is correct, then the algae are a carbon flux and not a source per se. Line 420: What does "reasonable" mean? Is this fit presented by the authors?

Line 428: What about heterogeneity of albedo? Albedo increases on a glacier as elevation is gained. What is the amount of variability in albedo for a surface that appears to be homogeneous?

Line 432: "brighten" should probably be changed to increases the albedo to that of fresh snow (∼0.85), before declining to seasonal normal values (over a given time period on the order of days).

Line 446: What does "some" mean in this instance?

Line 453: "reasonable" should be described. What is the difference between the two? Line 455: What is the temperature control on snow fall events? Snow does fall when the surface temperature is > 0.

Line 458: What does "this year" mean?

Line 465: Why were five realizations chosen?

Line 468: I don't remember seeing any run-off data?

Line 471: Please describe how this is a positive feedback. A warming atmosphere produces more rainfall events (instead of snow) at the glacier's elevation. Rain further melts the glacier causing more rainfall events?

Line 525: Are there no observations of this behaviour on Haig Glacier?

Line 529: From which transect date?

Line 536: This paragraph is mostly results and should be presented in the Results section. It is a bit problematic that the authors are relying heavily on unpublished data

to interpret the albedo results. Are the unpublished results necessary?

Line 551: Upon what basis is this statement made? There is no observation station at lower station, yet the melt feedbacks are the strongest here. What exactly is the melt feedbacks and why is this plural?

Line 578: Shouldn't start a new paragraph with "this".

Line 582: Does water vapour pressure increase over the study period, or for that matter, any of the other environmental variables measured at the weather station?

Line 590: What are the "ways" that you suggest?

Figure 4: Including dates for the photos would be helpful.

Figure 7a: No snow pits appear on the figure. The figure leads me to believe there are additional temperature measurements available.

Figure 8: The modelled values seem to reach a maximum at ∼0.85. What is the reason for this? The observed data clearly achieves higher albedo values.

Figure 9: There are ∼seven points in the above the trend line (f∼7; albedo∼0.7) that if removed would greatly improve the correlation. Can the author identify the origin of these points (i.e., a specific year, or month)?

---

## Author Comment (AC1) · 1 Jul 2020

Anonymous Referee #1 Review of: Seasonal and Interannual Variability of Melt-Season Albedo at Haig Glacier, Canadian Rocky Mountains Submitted to The Cryosphere by Marshall and Miller.

Jing Ming (Referee) petermingjing@hotmail.com Jing Ming Beacon Science & Consulting Melbourne, Australia Email address petermingjing@hotmail.com

The paper uses a long-term observation dataset of surface albedo in the Haig Glacier during the period 2002-2017 to depict the seasonal and Interannual Variability of Melt Season Albedo at Haig Glacier, Canadian Rocky Mountains. It is important to present this valuable dataset for developing any energy or mass balance model to project the evolution of the glacier. The tuning of the MB model is also a nice try. The paper is promising to be finally accepted by the Cryosphere from my point of view. However, before its formal acceptance, I want to address a few concerns here.

Thank you for your time and suggestions. We appreciate your thoughts on how to improve this manuscript. Please see below for our responses, in blue. Page and line numbers refer to the attached track-changes revised manuscript, not the TCD-formatted line numbering.

Specific comments:

1. The first paragraph of the Introduction part seems to describe the target of the work, which is more proper to be moved to the end of this part.

We rewrote this as suggested, and moved some of the specifics of this study (location, objectives) to later in the introduction.

2. Line 3-4. The sentence, "Variations in surface albedo, therefore, exert a strong control on the surface energy balance and available melt energy", needs a reference. Here is one by Ming et al. (2015) for your information. - Ming, J., et al. (2015). Widespread albedo decreasing and induced melting of Himalayan snow and ice in the early 21st century. PLoS One. 10: e0126235.

This and another reference from Oerlemans have been added here, p.2, l.14.

3. Line 4. "manuscript" -> "work" or "study".

Changed to "study" as suggested, p.3, l.23.

4. Line 44-51. This paragraph reads to be wordy and not well organized and needs to be rephrased.

We rewrote this to shorten some sentences and moved a bit of content to later in the introduction.

5. Line 44. The word "this" is not clear. Please clarify it.

"this" was deleted and we added "therefore", p.2, l.24.

6. Line 45-51. These two sentences are too long to read. Please rephrase them to several shorter sentences.

Rewritten; these sentences are now simplified, p.2, ll.25-30.

7. Line 97. Figure 1 had better incorporate a smaller map of the study area from a global perspective so that the readers could know where the study area is in the first sight. It is also beneficial to include the conditions of climatology for this area in the figure.

Sorry for the geographic assumption – we have added a larger map to indicate the area. We did not included climatology though – partly it is not known in the Rocky Mountains (e.g. monthly precipitation data is not really available, except in the valley bottoms where it is about 20% of what we measure on the glacier, based on the depth of the spring snowpack. Figure 1 has been revised to better indicate our study region within North America.

8. Line 97. "Albeta" -> "the Albeta province" or "the Albeta state" or "the Albeta city"?

Clarified: "provinces of British Columbia and Alberta", p.4, l.8.

9. Line 101-102. "Snow surveys conducted on the glacier each May indicate a mean winter snowpack of 1.35 m water equivalent (w.e.) on the glacier from 2002-2017, with a standard deviation (σ) of 0.24 m w.e. (Table 1)." Is this original from this study or cited from other studies? If it is in the latter case, it needs a reference. I suggest using a simpler expression of 1.35 ± 0.24 m w.e. to replace the long one in the previous form.

Revised as suggested, for the standard deviation, p.4, l.16. We needed to introduce/define this here, so it is a bit wordy. These numbers are newly reported in this study, a slight update from Marshall (2014).

10. Line 105. Could you also add a standard error of the mean of the temperature after the number 5.3 âDˇ C?

Added as suggested, p.4, l.19. Although it is not standard error, but rather the standard deviation (i.e., the interannual variability).

11. Line 111-115. This paragraph ˇ could be incorporated into the measurement section, and the next as well, because two paragraphs are more like introducing the measurement and data collection.

Shifted into Section 2.2 as suggested.

12. Line 116. "The forefield AWS" -> "The AWS in the forefield"? This phrase appears a few times throughout the text.

We think it is permitted to use "forefield" as an adjective, similar to "glacier AWS" or "forefield environment", but for clarity we have reworded this throughout the manuscript, e.g. p5, l.6.

13. Line 123. Please clarify what "the set of available in situ data" is.

We reworded this as well. We just mean the available data – the 79% of days from 2002-2015 with valid data (N = 1018), p.5, l.13.

14. Line 133-134. Here needs a more detailed description of how to do manual quality control and remove the questionable data, although the authors claimed that the data control had been introduced in Marshall (2014). The current is too simple to understand the method.

We added a few sentences to make this more self-contained, so that readers don't need to look this up elsewhere – thanks for this suggestion and we hope that it is now more clear, p.5, ll.3-8.

15. Line 135. "concentrates" -> "focuses" or "zooms in"? The usage of "concentrate" here seems to be strange.

Revised to "focuses on", p.5, l.29.

16. Line 135-136. The intent of the sentence is unclear, and please rephrase it.

Rewritten – we mean simply to define the variables and our notation here, p.5, l.30.

17. Line 136. "pragmatic" -> "virtual"?

Apologies, we have removed this word – it was unnecessary, p.5, l.31.

18. Line 137. "evolution" -> "variation"?

Clarified: seasonal evolution and interannual variation, p.5, l.31.

19. Line 142. "than" -> "from" or put "other" before it.

Revised to "from", p.5, l.38.

20. Line 150. Please clarify how you calculated out 7%.

Apologies, just from standard propagation of errors, and assuming 5% uncertainty in each of the incoming and outgoing radiation values: for z = x/y and uncertainties (dx, dy, dz),

dz/z = sqrt((dx/x)^2+(dy/y)^2) = sqrt(2*0.05^2) = 0.07

We add a short explanation, p.6, l.8, but don't include the equation in the text, as it is standard error analysis.

21. Line 157. The last sentence "modelling of potential reflected radiation from valleys walls indicates that this is negligible at our AWS site". Could you please present evidence of your claim?

This is a fair request. We have done the modelling as part of previous studies (Marshall, 2014; Ebrahimi and Marshall, 2016), but this specific result is not published and is ancillary to the focus of this study, so rather than include a Figure and the equations to explain this point, we have removed this sentence.

22. Line 159. "paper" -> "work". "repeat" -> "repetitive" or "repeated".

Revised to "study"; "repeat" deleted, p.6, l.20.

23. Line 161. "Haig Glacier albedo" -> "The albedo of the Haig Glacier".

Revised as suggested, p.6, ll.22-23.

24. Line 162. "points" (geometric concept) -> "sites" (geographic concept). Check that throughout the context.

Revised to "sites" as suggested. We consider it point data but it's true, we made multiple measurements over a few $m^2$, p.6, l.24.

25. Line 166. Was the sensor held manually? If so, how did you avoid the shadow of the body when measuring? Please clarify.

Yes, held manually, at arms length and pointed to the south to avoid shadows, p.6, l.33.

26. Line 167. Please give the detail of presuming a 10% uncertainty.

We added more detail on this. The manufacturer reported 5% accuracy, but we also observed fluctuations of a few 10s of $W/m^2$ while taking the readings of incoming shortwave radiation. e.g. for a value of 800 $W/m^2$, the instrument readings would bounce around between values of ~770 to 830 $W/m^2$. Readings of reflected shortwave radiation were much more stable. We therefore assign an additional 5% uncertainty in the observation itself, and add this to the instrumental accuracy to get what we consider to be a conservative estimate of 10%. Discussed on p.6, ll.34-38.

27. Line 169. "for melting and major ion and organic carbon analyses" -> "for the analysis of major ions and organic carbon". Please provide the source or references of the impurities used in this work.

These are detailed in Miller (2018), as cited. We are preparing a separate manuscript examining the impurities in detail, but with much of this beyond the focus of this study. That said, we recognize the importance of having the essential data that we refer to presented within this study, so we have added these results (Table 5) as well as essential details on the sampling and analysis, p.7, ll.4-14.

28. Line 176. "data" -> "temperature" and "precipitation"? Please specify them.

It is an energy balance model, so the full suite of meteorological data as described earlier. We now list this explicitly, p.4, ll.31-33.

29. Line 177. Please check the use of articles throughout the context. "forefield AWS data" -> "the data from the AWS in the forefield".

We deleted this part as it was redundant from the QC and gap-filling explanation in Section 2.2.

30. Line 191. What do you mean "the net energy goes to melting"? Please rephrase it.

We rephrased this as requested, p.7, l.35 – we mean that the energy is directed to melting.

31. Line 195. Give out the exact value of Lf (334 J g−1).

We don't systematically note the values of all of the established constants that are used in the energy balance model, but for clarity we added this here, as well as the density of water, p.7, l.39. Both values are standard but this does not distract too much from the flow of the narrative.

32. Line 240. Please clarify the definitions of a and b, respectively.

Regression coefficients – now defined, p.9, l.9.

33. Line 430. The first sentence needs to be rephrased. Do you mean "the impact of fresh snow on albedo"?

Rephrased for clarity, p.15, ll.18-20.

34. Line 450. "forced" –> "driven".

We see these as interchangeable in common usage, but revised to "driven" as suggested, p.16, l.7.

35. Table 1. Please clarify the definitions of summer and winter for this study in the caption or context.

This is now added to the text in Section 3.1, as the caption is already long and wordy. Our definitions are conventional for mid-latitude glaciers: winter accumulation is from the end of the previous melt season to the subsequent spring (i.e. the start of the next melt season), so roughly October to May at our site. Summer, glaciologically, refers to the melt season, roughly May to September at our site. The exact days vary from year-to-year and over the glacier.

36. Figure 2. Why didn't the authors use the lines of means with shaded area indicating the error?

We think the reviewer is asking for a plot that includes the standard deviation of the measurements? We have added this in Figure 2a, but will leave Figure 2b as is to avoid clutter. Our intent with this plot was not to show the errors but rather then mean and minimum values associated with the 14-year observational record.  i.e. the minimum here is not an error, but the lowest daily mean value recorded for that day over the 14 years. We have retained that, as it gives a clear indication of the "bare ice" season.  But the inclusion of a shaded region for $\pm 1\sigma$ is useful additional information. Note that if the reviewer was actually requesting error bars, this is not what we have added here. These are very small for the average daily values: with an uncertainty of 7% in the mean daily albedo, the average over 14 years has an associated uncertainty of about 2%. (i.e. Or to be explicit from the quadrature rule for error propagation, for an example with $\alpha_s = 0.60 \pm 0.04$, we have $d\alpha_s = 0.04$ and N = 14. The error in the mean is $d\alpha_s/\sqrt{N} = 0.01$.)

37. Figure 7. The blue points denoting the snowpits are blur.

Thank you – we revised this to make them clear.

38. Figure 8. What about the significances between the observed and modelled?

It is inappropriate to compare the daily modelled vs. observed time series statistically, e.g. for correlation or $R^2$, as the stochastic model does not attempt to resolve the exact timing of specific snow events. This is a bit like weather vs. climate modelling. Our aim is not to represent a specific day, but rather the mean summer albedo value and the general seasonal evolution. The mean values can be compared through a standard t-test, and the observed vs. modelled variance can be compared with Bartlett's test. The statistical tests indicate that the mean and variance are statistically equivalent (p > 0.001). We now report this, p.16, ll.21-25.

39. Figure 9. The same issue as that in Figure 8. Significance?

We now add the $R^2$ value and note the significance of the linear relation, p.17, ll.23-25. Good suggestion, thank you.

40. The language of the context needs a thorough check for grammar and misused words, such as articles, the function word "of", ambiguous statements, etc.

We have read and edited carefully and believe that the text is in proper and clear English, but we welcome any additional specific comments where our writing is ambiguous.

---

## Author Comment (AC3) · 1 Jul 2020

Review of: Seasonal and Interannual Variability of Melt-Season Albedo at Haig Glacier, Canadian Rocky Mountains Submitted to The Crysphere by Marshall and Miller.

Major Revisions required.

Albedo measurements from in-situ weather stations are used to identify melt season albedo dynamics for Haig Glacier. The results are used to comment on the conventional application of degree-day melt rates and on how albedo describes glacier mass balance. C1 TCD Interactive comment Printer-friendly version Discussion paper These types of in-situ data driven papers are very important to the understanding of glacier dynamics and glacier mass balance, especially for mountain glaciers.

AU: We thank you for the time spent reviewing the manuscript and providing constructive suggestions for improvement. These are all helpful suggestions and we believe that we have been able to respond to these, leading to a better-organized and more clear contribution. Please see below for our point-by-point response, in blue. Page and line numbers refer to the track-changes copy of the manuscript.

The manuscript is well written with a logical presentation of material. I would suggest a minor re-organisation of the Introduction section to separate the literature review from specific mention of the study on Haig Glacier, as the sporadic reference to the study on Haig Glacier comes across as a bit disjointed. The final paragraph of the Introduction should be devoted to specific details regarding Haig Glacier. Specifically, how the study on Haig Glacier addresses the limitations related to glacier albedo and modelling.

We agree, we were jumping around far too much in an attempt to state the objectives of the paper in the opening paragraph. We have now reorganized as suggested, with the specific details of the study site and the aim(s) of the study in the final two paragraphs.

Abstract: The improvements related to the stochastic model on mass balance and the modification of the degree day model should be provided in more detail.

This is difficult with the limited space, but we have revised and added more detail here. This may need to be trimmed in the next round of revisions, as we are at 389 words for the abstract.

Line 11: Summer should be defined in the abstract (e.g., June 1 to August 31). Summer is defined on Line 104.

Thanks – JJA is now defined in the abstract.

Body of text:

Line 28: It is true that albedo is involved in the control of surface energy balance, but it is the net radiation (short wave and long wave) that mostly controls melt. Net radiation was previously mentioned, but a better description of how net radiation is related to albedo and what the proportion of shortwave to longwave radiation is, would be very useful.

This is true – it is net radiation that really matters, but with albedo as the main influence on melt-season variations in net radiation on mid-latitude mountain glaciers. It is a bit hard to compare the importance

of net SW and net LW balances, as the latter is an energy sink. Hence we cannot say that X% of the melt energy is due to absorbed shortwave radiation and Y% from the net longwave. As a measure of this, we now report the correlation of each to the net energy that is available for melt, based on previously published data at our study site (Marshall, 2014). This interferes with the attempt to move all mention of Haig Glacier to the end of the introduction (per below), but it is relevant here and addresses this request to articulate the importance of albedo. Other references to Haig Glacier have been moved to the end of the introduction, as suggested.

We calculate the mean daily surface energy fluxes for the set of all summer (JJA) days reported in Marshall (2014), $N$ = 1012. The average net energy, $Q_N$, is 101 W/m$^2$, with 79 W/m$^2$ from the net radiation, Q*, and 22 W/m$^2$ from the turbulent fluxes (26 W/m$^2$ from the sensible heat flux, $Q_H$, and −4 W/m$^2$ for the latent heat flux, $Q_E$). Net radiation accounts for 79% of the net energy that is available for melt. Within this, net radiation is dominated by net shortwave radiation in the summer months: 107 W/m$^2$, vs. −28 W/m$^2$ for the net longwave radiation. We also calculate Pearson's linear correlation coefficients, $r$, for net energy against each of the radiative fluxes and the albedo ($N$ = 1012):

$r$ ($Q_N$, SW in) = 0.39
$r$ ($Q_N$, net SW) = **0.84**
$r$ ($Q_N$, albedo) = **−0.81**
$r$ ($Q_N$, LW in) = −0.09
$r$ ($Q_N$, net LW) = −0.20
$r$ ($Q_N$, net radiation) = **0.94**        $r$ (net radiation, albedo) = **−0.83**

These values are summarized in the introduction, although we tried not to get too bogged down in what feels like results (albeit from previously published data), p.2, ll.6-12. We also rewrote this to clarify that net radiation dominates net energy, but net shortwave radiation dominates net radiation in the summer melt season (with albedo being the main control of daily mean net shortwave radiation).

Line 29-30: Reference to Haig Glacier should probably come at the end of the introduction.

This sentence has been moved to the end as suggested.

Line 48-51: This sentence seems to be a bit misplaced and should be moved to the end of the introduction as a bridge between the literature review and the methods section.

True, our apologies. This was definitely out of place. Now moved to later in the introduction.

Line 64: Please define what a melt-albedo feedback is.

We added a sentence to explain this positive feedback, p.3, ll.8-9.

Line 69-71: Snow algae can be of many species (up to 4 or 5). Is there a reference for this material, or is it an observation from Haig Glacier? If it is an observation it would find a better home in the Results section.

This is just an observation from Haig Glacier, a common spring occurrence. In fact we don't know the species for certain, though I have been told it was *Chlamydomonas nivalis.* This comment was meant to make the reading more interesting but is not needed, so we have removed it in the event that we have the wrong species of 'pink algae'.

Line 76: A recent article in Remote Sensing of Environment might be of interest here: Williamson et al., 2020 - Comparing simple albedo scaling methods for estimating Arctic glacier mass balance.

Thank you – now cited, p.3, l.21.  Happy to have this paper brought to our attention.

Line 78-79: This material might be better suited in the final paragraph of the Introduction.

Thanks, also moved in the rewrite.

Line 82: Can you provide more detail on what the "simplified parameterizations" entails?

We now refer to these explicitly as temperature index melt models, described in more detail in the abstract and in the lines above and below the sentence that was flagged, p.3, ll.33-41.

Line 111: Campbell does not make many instruments. The details for the instruments should be included (manufacturer and instrument), at the very least for the radiometric instruments, as different instruments are sensitive to different range of the EM spectrum.

This is a good request, for a paper focused on albedo. This information has been added, p.4, ll.36-42.

Line 119-120: Data collection ongoing has previously been mentioned.

Thanks, now deleted.

Line 124: If only one station is collecting data how was the lapse rate estimated? Please provide details.

Details now provided, p.5, ll.16-26. This is based on the 'climatological' mean lapse rates at this site (or really just offsets, with two points), calculated from the multi-year record for all days when both stations were working. This gives daily and monthly mean values for the offset, or one can calculate lapse rates from this for glacier-wide application.

Line 126: How much error is related to the estimation?

This is a good question. Where forefield data are available, which covers about 70% of the data gaps, error is small because we understand the relation well between the forefield and glacier AWS records. The stations are 2.5 km apart, although there are systematic (and seasonally-varying) offsets associated with the different environments: snow/ice vs. rock.  Where both stations are missing data, we fill in with the average value for that day from the 'climatological' (historical) data for that day, the mean of available data from 2002-2015. The error can be quantified by applying the gap-filling procedure to estimate data for times with valid data. For interest: comparing observed temperature at the AWS site (as an example) to adjusted AWS data from the forefield gives an average error of $-0.13°C$ (a small cold bias), while using the 'climatological' mean value gives an average error of $-0.11°C$.  A similar analysis for specific humidity gives values of 0.15 g/kg and 0.16 g/kg, compared with a mean value of 3.3 g/kg: hence an error of 5%. These values are for the 30-minute data.

We don't present this because we don't use gap-filled data for the albedo values that are reported here (cf. p.5, ll.9-10) – only the days with quality-controlled in situ data are used for the albedo statistics and plots. That is the primary focus of this study. We do use the gap-filled data to drive the surface energy balance model, e.g., to evaluate the sensitivity of modelled melt to albedo.  This is secondary to the main results and discussion, however. A formal error analysis could be done to propagate the error in

temperature, wind speed, etc. through the surface energy balance equations, but this would be a tangent to the main points of the manuscript. Interestingly, I seldom see this in surface energy balance studies, i.e. assessment of uncertainty in the meteorological forcing and how this propagates through to errors in the surface energy fluxes.

Line 132: Define "questionable data".

We expand on this now, p.5, ll.4-6 – physically impossible values, off-scale readings (-6999), and 'flat-lining' that we sometimes see if a sensor gets buried by snow in the winter.

Line 154: There is a recursive reflection from the bottom of optically thin clouds or from scattered clouds and a high albedo snow covered surface.

Yes, interesting, but this should be implicitly accounted for in the radiation measurements. The incoming radiation sensor would measure this reflection from the clouds and it would be twice-reflected from the glacier surface. This can lead to overestimates of both the incoming and outgoing shortwave radiation, but this should scale without major effects on the albedo. Small effects are possible by changes in the composition of diffuse vs. direct radiation, but we do not separate these in this study.  As a side note, we did examine subsets of overcast vs. clear-sky days, and found no statistically significant differences in average snow or ice albedo on the glacier on these days.

Line 165: Please define Jaycar QM1582. What is the spectral range of this instrument?

This is just the brand name of the specific handheld pyranometer we used. Thanks – we now report the spectral range, which does differ from the Kipp and Zonen instruments. Caution is therefore needed in comparing these values with the AWS albedo records, but within the particular spatial surveys conducted in 2017, we can compare these values in space and in time (i.e. for the four repeat surveys). We add a note of caution on comparing with the AWS-measured broadband albedo, p.6, l.31 to p.7, l.2.

Line 221: "this" should be these.

Revised as suggested, p.8, l.27.

Line 245: The introduction mentions two AWS. It is not clear which station these results refer to. I assume from the data period this is the on ice station (upper ablation zone).

Sorry yes, all of the albedo results are from the glacier AWS – the off-glacier AWS is not helpful here, but is just used in this study for gap-filling of missing meteorological data for the energy balance modelling. We clarify here, p.9, l.17.

Line 284: "jump" might not be the best descriptor here.

We revised this to "increase", p.10, l.29.

In Table 2 why is Em larger for August than July? Cloud cover – because Em is using only shortwave radiation?

No, Em also includes longwave radiation.  All of the terms in the energy balance, per equation (1). Cloud cover is not the cause - it is in fact directly due to the lower surface albedo in August.  Much more shortwave radiation is absorbed in August than in June or July.

Line 289: What type of regressions are these? Linear, least-square regressions, Pearson's? Are the correlations statistically significant? If so, which ones?

These are simple linear Pearson's correlation coefficients. Now stated. We also now indicate in the Table which values are statistically insignificant (p > 0.05).

Line 291: What does "correlated" mean in this instance?

Here we mean to say there is a statistically significant negative or positive correlation. This should now be clear from the explicit indication of this in Table 3. The discussion on pp.10-11, Section 3.2, has been revised accordingly.

Line 297: Define "fewer samples".

We specified the numbers but have rewritten through here: we have 14 years of data, 2002-2015, so N=14 for the mean summer conditions and their relation to the annual mass balance conditions. Within each year we analyze data from May through September, giving us 70 months. This sentence has been removed in place of a more clear discussion of sample size, p.10, ll.40-43.

Line 300: Define "melt out" or replace with better descriptor.

Revised to the more specific/technical term "ablate", p.11, l.13.

Line 309: Define "ripened and saturated"?

Revised to "wet, temperate" (at 0°C, with liquid water content), p.11, l.28.

Line 315: Some other citations that might be useful here, especially in the context of spatial variability of albedo. 1. B.W. Brock, I.C. Willis, M.J. Sharp. Measurements and parameterization of albedo variations at Haut Glacier d'Arolla Switz. J. Glaciol., 46 (2000), pp. 675-688 2. S.N. Williamson, L. Copland, D.S. Hik. The accuracy of satellite-derived albedo for northern alpine and glaciated land covers Polar Sci., 10 (2016), pp. 262-269

We were already citing the Brock et al. (2000) paper here, p.11, l.37. We prefer to stay with comparisons to direct/in situ, broadband albedo measurements here, but the Williamson et al. (2016) paper is very relevant to later sections where we discuss spatial variations and satellite measurements of albedo, so we have added this there, p.17, l.12.

Line 323: Describe the film, thickness composition, etc. Is there liquid water in the surface matrix? If so, what effect does this have on albedo? O.k., I see this is addressed on Line 335.

It's about a 1 mm film, with examples in Figure 4, although it is not a continuous film everywhere – in many places impurities are discrete particles, with varying density. Now noted, p.12, l.4. Like most mid-latitude mountain glaciers, the glacier surface is wetted during the summer melt season, but well-drained. Certainly these two effects – the impurities and wetness – contribute to the low values of ice albedo, as discussed, and the generally lower albedo of mountain glaciers compared to polar ice.

Line 325: Not clear where the values for Figure 5 are coming from, and provide how N=224 was derived.

This is for all bare-ice days in the 14-year record (i.e. when there was snow cover at the AWS site). N=224 is the number of days, derived by counting all days with albedo values less than 0.4 after the

initial rapid drop in albedo (the snow to ice transition) that is clearly evident each summer (e.g., Figure 3b). We have edited to clarify this, p.11, l.31. The caption of Figure 5 is also revised.

Line 343: Adding year to the dates will reduce confusion.

We note the year now in introducing this discussion, p.12, l.27.

Figure 6: Mean values should have standard error included on the figure.

We added this for plot 6a. The mean multi-year values have very low standard errors: for a mean daily error of 7% and for 14 years, averaging reduces this to about 2%. Standard error (the uncertainty envelope, really), is higher for individual years, as plotted for the data from 2003 in Figure 6a. For Figure 6b, the mean daily value for each year, we don't plot this because errors are vanishingly small – these values are calculated from a mean over either 92 days (JJA) or 152 days (MJJAS) for each year. On averaging, the uncertainty in an error $d\alpha$ (7% for a mean daily value) is calculated from $d\alpha/sqrt(N)$, so for JJA this is 0.7%, or 0.004 for an albedo value of 0.6. This is not easily visible on the plot. Note that we have interpreted this request as the reviewer's suggestion to plot the standard error where possible and relevant – we assume that the reviewer is referring to standard error, not standard deviation.

Line 376: "dropping" should be decreasing.

Revised as suggested, p.13, l.29.

Line 385: The values of ~0.1 and 0.07 are close enough that instrument error might render these inseparable?

We conservatively estimate the instrument error to be 10%, double that of the manufacturer-specified accuracy. By taking the average of three measurements, this is further reduced, to 8%. But even at 10%, this means 0.10 ± 0.01 and 0.07 ± 0.007: 0.10 and 0.07 are statistically distinct. We also measured the 7% albedo at 3 different sites (the lowest three points) on the centreline transect.

Line 395: Can evidence be presented that Haig glacier was indeed downwind of the forest fire smoke? For example, can specific fire events be linked to specific albedo declines for 2017? Without this link the material presented here is speculation.

In fact we also consider this to be speculation here, and tried to phrase it that way. That said, we were up on the glacier in thick smoke for many days (smelling of smoke, hazy skies with limited visibility, direct observations of it blowing in from the southwest). Winds on the glacier systematically blow in from the southwest (B.C.), funneled by the valley geometry. We also have wind direction data to support this. However, a thorough analysis of specific forest fire events, black/organic carbon provenance, and plume modelling is beyond the scope and focus of this study. We comment on this explicitly now, and make it more explicit that "we speculate…", p.14, l.9, and subsequent lines. We also note our direct observations of forest fire impacts during this period, as well as the indirect evidence through the increase in particulate concentrations, now included in Table 5.

Line 399: Which year?

2017, per this entire section – now noted

Line 400: Please present pertinent details for the data. The reader can't evaluate the data from an unpublished source.

This is a valid request – apologies not to include this earlier. We had cited the MSc Thesis of Miller (2018), which is available online and contains all of the details, but agree that it is helpful if the manuscript stands alone. We now present the data that we refer to in Table 5. Interested readers can find more detailed data tables and analyses in Miller (2018), but we now include the referenced data in this study.  Additional supraglacial and meltwater chemistry data in Miller (2018) will support a separate publication on the supraglacial chemistry of Haig Glacier and its evolution through a melt season. As much of this is not essential and is ancillary to the current study, we present only the data that shows the large increases in impurities and carbon concentrations through the period of regional forest-fire activity in summer 2017, coincident with the observed decrease in ice albedo through this period.

Line 401: How is this "consistent"? Provide details, references or rationale.

Consistent in that forest-fire fallout would be expected to be carbon-rich, so-called 'brown carbon' as well as black carbon and soot (e.g., C.J. Williamson et al., 2020). But as this is results and not discussion, we removed this comment and now just present the observations and data, without commentary.

Line 407: I assume that algae assimilate carbon that was on the glacier before, or during, its growth. If this is correct, then the algae are a carbon flux and not a source per se.

This is partially true – they assimilate available nutrients – but they are also autotrophic, prolifically photosynthesizing and engaging in atmospheric carbon fixation. See e.g. C.J. Williamson et al. (2019), cited in the manuscript, as well as Yallop et al. (2012), Cook et al. (2012).

Yallop, M.L. *et al, 2012.* Photophysiology and albedo-changing potential of the ice algal community on the surface of the Greenland ice sheet. ISME J., 6, 2302–2313.

Cook, J. M., Hodson, A. J., Anesio, A. M., Hanna, E., Yallop, M., Stibal, M., et al. (2012). An improved estimate of microbially mediated carbon fluxes from the Greenland ice sheet. *J. Glaciol.* 58, 1098–1108. doi: 10.3189/2012JoG12J001

Line 420: What does "reasonable" mean? Is this fit presented by the authors?

Good point, this is imprecise language. We do present statistical fits below, after the introduction of stochastic snowfall events. But for this occurrence, we have revised this sentence to remove this statement, p.15, l.5.

Line 428: What about heterogeneity of albedo? Albedo increases on a glacier as elevation is gained. What is the amount of variability in albedo for a surface that appears to be homogeneous?

We consider variations with elevation in sections 3.4 and 4.3. The albedo increase with elevation is not generally observed when it is all snow-covered (i.e. for up to 9 months per year in the Rockies), but is true in the summer melt season when lower elevations have exposed ice (e.g., Figure 7b). Also, for the exposed glacier ice itself, albedo increases with elevation have been reported elsewhere and are seen on Haig Glacier as well. This is associated with increasing concentration of impurities at lower elevations, and could be incorporated in Eq. (7) through the value of $k$ if one had an idea of the spatial variability of

impurities and their influence on snow albedo. But Eq. (7) does not refer to glacier ice, which is where impurity-driven spatial heterogeneity has been documents. Applied across a glacier, Eq. (7) does capture differing rates of melt (i.e. greater PDD at lower elevations) and these effects on albedo decline (wetting, recrystallization, the timing of the transition from snow to ice).

Line 432: "brighten" should probably be changed to increases the albedo to that of fresh snow (~0.85), before declining to seasonal normal values (over a given time period on the order of days).

Revised to remove 'brighten' and use the language 'increase the surface albedo', p.15, l.19

Line 446: What does "some" mean in this instance?

Some has been removed, p.16, l.2 – you are right, not meaningful here. We also added a sentence to explain this more clearly as well, p.16, ll. 2-3 (i.e. random sampling of a normal distribution to introduce realistic variability in this value, vs. assigning a single value).

Line 453: "reasonable" should be described. What is the difference between the two?

Thanks – this was imprecise language again. We have revised to "is in accord with the observations", p.16, l.10. We also statistically assess this now, per the comments of both reviewers, p.16, ll.20-25.

Line 455: What is the temperature control on snow fall events? Snow does fall when the surface temperature is > 0.

Yes, recognized, as it the air temperature (the column) and not the surface temperature that matters. Rain also falls at near-surface temperatures below 0°C. Snow is increasingly unlikely as temperatures increase, however, so we parameterize this simply based on a linearly-decreasing snow fraction $fs$ from 1 to 0 between near-surface air temperatures $Ta$ of $-2$ to 2°C. This uses mean daily temperatures. Now explained, Eq. (8), p.15, l.34.

Line 458: What does "this year" mean?

The year being discussed and plotted here, 2007 – now stated again on p.16, l.19.

Line 465: Why were five realizations chosen?

This was arbitrary. In putting together our statistics we increased this to 10 realizations. Each one differs a bit (e.g. Figure 7b), so it is better to include several realizations in the mean, but the values after just a few realizations converge and are representative of model results for a given set of parameters.

Line 468: I don't remember seeing any run-off data?

True, we don't report that here. Like most mountain glaciers, all summer ablation runs off, based on past studies and discharge measurements at this site, so we commonly equate these. But to be careful here, we have removed the reference to runoff, p.16, l.38.

Line 471: Please describe how this is a positive feedback. A warming atmosphere produces more rainfall events (instead of snow) at the glacier's elevation. Rain further melts the glacier causing more rainfall events?

Thank you, good catch. Indeed this is not a feedback, although it excited albedo-melt feedbacks on the glacier. Wording changed that this would accelerate the melting, but of course without feeding back on the precipitation, p.16, l.42.

Line 525: Are there no observations of this behaviour on Haig Glacier?

To our knowledge, glacier albedo trends have not been detected or reported on Haig Glacier or in the Canadian Rockies. Only anecdotal impressions. One of our objectives in this study was to analyze this from our long-term observations, to provide the first assessment of whether albedo is declining. Happily for the glacier, we don't see any evidence of albedo declines over the study period, so we have to reject our null hypothesis that the glacier is darkening due to an extended period of negative mass balance.

Line 529: From which transect date?

This is from late summer, when the seasonal snow is gone and we are comparing ice with ice. Now clarified, p.18, l.28.

Line 536: This paragraph is mostly results and should be presented in the Results section. It is a bit problematic that the authors are relying heavily on unpublished data to interpret the albedo results. Are the unpublished results necessary?

Thank you, good comments. As discussed above, we had referenced Miller (2018) for the data, which is published on line and peer-reviewed, insofar as graduate theses have been vetted. But we agree that it is better to have the data presented here, so it is now included in Table 5 and discussed in the results. In the context here, we are going beyond results and talking about the potential for melt-induced concentration increases (vs. atmospheric deposition) – it is more discussion and interpretation than results. We have revised this paragraph though, to refer back to Table 5 rather than present new numbers/results here, p.18, l.37 to p.19, l.4.

Line 551: Upon what basis is this statement made? There is no observation station at lower station, yet the melt feedbacks are the strongest here. What exactly is the melt feedbacks and why is this plural?

Thanks, this was unclear as written. By "changes and melt feedbacks have been strongest here", we were referring to the mass balance and glacier thinning over the study period. The toe of the glacier has largely collapsed.  But we don't have albedo data to comment on changes in albedo or whether the lower ablation zone is getting darker. We have rewritten this to specify that we mean "where glacier thinning and mass loss have been most extensive", p.19, ll14-15. We removed the discussion of melt feedbacks – plural because there are a few things happening, e.g. lower elevation = warmer and more melt; more exposed bedrock warms up and melts the glacier terminus more, though sensible and longwave fluxes; potential accumulation of dust and debris which warms up and melts the glacier more, etc.  These are all known processes but we don't measure them or present data on these in this manuscript, so we have taken this out.

Line 578: Shouldn't start a new paragraph with "this".

Revised as suggested.

Line 582: Does water vapour pressure increase over the study period, or for that matter, any of the other environmental variables measured at the weather station?

We did not analyze this here, so won't introduce it in the conclusions and will keep the focus on the albedo measurements and modelling. There are increasing trends in summer temperature and melting/mass loss, though with a lot of interannual variability and so only weakly significant. There is no statistically significant trend in vapour pressure. This question concerns the glacier mass balance and weather trends, which are not the subject of this study, so we don't add this to the manuscript in the interests of keeping our focus. At this particular line, we are discussing how summer snowfalls (in general) impact the mass balance, not the trends in such events or in mass balance.

Line 590: What are the "ways" that you suggest?

Apologies, this would only be clear for those that read the results and discussion – it should be explicit in the conclusion (monthly factors or as a function of albedo), now stated, p.20, ll.21-23.

Figure 4: Including dates for the photos would be helpful.

To be honest, we don't know the exact dates, but also that is not important to the visual context that we wish to convey.

Figure 7a: No snow pits appear on the figure. The figure leads me to believe there are additional temperature measurements available.

We have revised the figure to better show the snowpits. It's true, there were three additional weather stations on the glacier this summer (Veriteq/Vaisala temperature-humidity stations), but we don't refer to these data so we have removed this from the plot.

Figure 8: The modelled values seem to reach a maximum at ~0.85. What is the reason for this? The observed data clearly achieves higher albedo values.

This is true – we set a maximum fresh-snow albedo of 0.85 in the summer snowfall model (as defined on p.15, l.10), but this is a free parameter that could be between ~0.75 and 0.9, looking at the data from Figure 8 as pointed out. Most of the fresh-snow events in August and the first two weeks of September that year are experienced by a rapid increase in albedo to values close to 0.8, whereas May snow events come in closer to 0.9.  Our value, 0.85, is taken as an average fresh-snow value, not the maximum.

Figure 9: There are ~seven points in the above the trend line (f~7; albedo~0.7) that if removed would greatly improve the correlation. Can the author identify the origin of these points (i.e., a specific year, or month)?

This is interesting and we had thought about this, but cannot justify removing these points. Four of these occur in May, of various years.  One is from June, two are from September. We don't observe anything special about these specific months, in terms of the temperature or other aspects of the meteorological conditions. These datapoints imply that there are certain times where there are high rates of melting per PDD even with high-albedo snow cover – degree-day factors of 7 are more typical of ice. A plot of just JJA conditions would give a stronger regression, but melt modelling needs to be inclusive of the shoulder season, May and September (e.g. melt totals in Table 2), so we retain these points, but can't explain what was different about these specific months. There is a fair amount of scatter at all albedo values in Figure 9 – the relation is significant but not as strong as we had hypothesized it would be ($R^2$ = 44, i.e. albedo explains only 44% of the variance in the melt factor, as discussed on p.17, ll.22-25).

---

## Author Comment (AC4) · 28 Jul 2020

The meteorological data used in this study are available at: https://doi.org/10.5683/SP2/7CXPPI

Marshall, S.: Daily meteorological data at Haig Glacier, Canadian Rocky Mountains. Scholars Portal Dataverse, doi:10.5683/SP2/7CXPPI, 2020.